# Magnetic Nanomaterials as Catalysts for Syngas Production and Conversion

Natarajan Chidhambaram [1,*], Samuel Jasmine Jecintha Kay [1], Saravanan Priyadharshini [1], Rajakantham Meenakshi [2], Pandurengan Sakthivel [3], Shanmugasundar Dhanbalan [4], Shajahan Shanavas [5,6], Sathish-Kumar Kamaraj [7] and Arun Thirumurugan [8,*]

1. Department of Physics, Rajah Serfoji Government College (Autonomous), Thanjavur 613005, India
2. PG and Research Department of Physics, Cauvery College for Women (Autonomous), Tiruchirappalli 620018, India
3. Centre for Materials Science, Department of Physics, Faculty of Engineering, Karpagam Academy of Higher Education, Coimbatore 641021, India
4. Functional Materials and Microsystems Research Group, RMIT University, Melbourne, VIC 3000, Australia
5. Department of Chemistry, Khalifa University of Science and Technology, Abu Dhabi P.O. Box 127788, United Arab Emirates
6. Department of Conservative Dentistry and Endodontics, Saveetha Dental College and Hospitals, SIMATS, Chennai 600077, India
7. Instituto Politécnico Nacional (IPN)—Centro de Investigación en Ciencia Aplicada y Tecnología Avanzada, Unidad Altamira (CICATA-Altamira), Carretera Tampico-Puerto Industrial Altamira Km 14.5, C. Manzano, Industrial Altamira, Altamira 89600, Tamaulipas, Mexico
8. Sede Vallenar, Universidad de Atacama, Costanera #105, Vallenar 1612178, Chile
* Correspondence: nchidambaraselvan@gmail.com (N.C.); arunthiruvbm@gmail.com (A.T.)

**Abstract:** The conversion of diverse non-petroleum carbon elements, such as coal, biomass, natural/shale gas, and even $CO_2$, into cleaner hydrocarbon fuels and useful chemicals relies heavily on syngas, which is a combination of CO and $H_2$. Syngas conversions, which have been around for almost a century, will probably become even more important in the production of energy and chemicals due to the rising need for liquid fuels and chemical components derived from sources of carbon other than crude oil. Although a number of syngas-based technologies, including the production of methanol, Fischer–Tropsch (FT) synthesis, and carbonylation, have been industrialized, there is still a great need for new catalysts with enhanced activity and adjustable product selectivity. New novel materials or different combinations of materials have been investigated to utilize the synergistic effect of these materials in an effective way. Magnetic materials are among the materials with magnetic properties, which provide them with extra physical characteristics compared to other carbon-based or conventional materials. Moreover, the separation of magnetic materials after the completion of a specific application could be easily performed with a magnetic separation process. In this review, we discuss the synthesis processes of various magnetic nanomaterials and their composites, which could be utilized as catalysts for syngas production and conversion. It is reported that applying an external magnetic field could influence the outcomes of any applications of magnetic nanomaterials. Here, the possible influence of the magnetic characteristics of magnetic nanomaterials with an external magnetic field is also discussed.

**Keywords:** magnetic materials; catalysts; magnetic separation; syngas; $CO_2$ conversion

## 1. Introduction

There has been a lot of scientific interest in the production of syngas ($H_2$ + CO) from biomass by gasification, a thermochemical conversion process that occurs in an oxygen-lean environment [1–3]. Syngas offers an effective alternative as a fuel for transferring energy, a raw material for creating hydrogen fuel, and a source of higher-value chemicals. Through more effective syngas-based power generation facilities and fuel cell technologies, syngas

production significantly reduces greenhouse gas emissions, particularly carbon dioxide ($CO_2$) emissions [4]. Nevertheless, it is thought that precious metal-based catalysts are the best option for producing syngas. However, significant limitations, including massive cost and unavailability, will eventually prevent their widespread use. Because of this, new non-precious metal materials have been developed as substitutes [5]. By enabling reagent adsorption and surface reactions on magnetized Fe catalysts as well as non-magnetic materials, magnetic fields can significantly improve these metals' catalytic activity. This improves selectivity to hydrocarbons and lowers apparent activation energy [6]. Although there are various metals used for syngas production using magnetic effects, Fe, Ni, and Co metals play a significant role in the production of syngas. Additionally, they have been used to generate electrically conductive materials and enhance the mechanical performance of composites. The qualities of magnetic nanoparticles include their capacity to transport other substances, their quantum characteristics, and their high surface-area-to-volume ratio. Magnetic fields can be used to alter the characteristics of magnetic nanoparticles to make them appropriate for a variety of applications, including syngas production, on a large scale. The efficiency of magnetic fields relies on the field gradient and particle magnetic moment. Based on the material, the optimum magnetic nanoparticles are between 10 and 20 nm in size as, above a certain temperature known as the blocking temperature, these particle unite into a single domain and show superparamagnetic activity [7]. However, this also leads to intrinsic instabilities over extended periods and magnetic loss because of chemically highly active bare metallic nanoparticles. The particularly attractive characteristics of spherical and cubic magnetic nanoparticles have attracted a lot of interest. It is noteworthy that these nanoparticles have been efficiently used in the pharmaceutical field for drug delivery, high-contrast MRI, DNA detection, and stem cell labeling/separation. In an initiative aimed at improving the catalytic efficiency in syngas production by using magnetic nanoparticles, it has also shown an efficient effect. This review discusses recent progress made on the effective utilization of magnetic nanomaterials for syngas production/conversion and the effective way of utilizing an external magnetic field for the synthesis of magnetic catalysts and for the adjustment of the efficiency of the synthesis/conversion process.

## 2. Magnetic Materials for Syngas Production/Conversion

Magnetic nanomaterials are found to be interesting in various applications due to their unique magnetic response behavior compared to other available nanomaterials. These materials are generally classified as magnetic and non-magnetic materials. Furthermore, they have sub-classifications, including dia, para, ferro, ferri, and antiferro, depending on the magnetic characteristics. Depending on the shape and size, their magnetic behavior is further classified as soft and hard magnetic materials. As a catalyst, transition metals are considered a support substance for the generation of syngas because of their mechanical, electrochemical, and metallic-like qualities. This may change a precious metal overlayer's catalytic characteristics. Transition metals had been synthesized using a nitriding technique that was carried out in the absence of hydrocarbons and only produced residual physisorbed N that easily desorbed as $N_2$ off the catalyst surface, in contrast to the situations of carbide and sulfide-based substrates [8]. When selecting active metals for the conversion of syngas into various sorts of products, we may be able to use the ability of CO dissociative adsorption as a guide. Due to their low adsorption heat, transition metals on the right of the periodic chart, including Cu and Pd, are not CO dissociative [9]. With the C-O bond uncleaved during CO hydrogenation, Cu and Pd are quite well-active phases for the production of methanol. However, the transition metals on the left of the periodic table, including Mo and W, exhibit a strong capacity for CO dissociative adsorption and the formation of metal carbides [10]. In contrast to Mo and W metals, Mo and W carbides are capable of activating CO differently and exhibit CO hydrogenation activity. Fe, Co, and Ru are appropriate for CO hydrogenation (Sabatier's principle) due to the moderate efficiency of CO adsorption on these metals. Rh is a promising candidate for ethanol formation from syngas because it is situated at the boundary of the CO dissociation and non-dissociation

regions. Notably, the balance between CO and $H_2$ dissociation capacities plays a crucial role in defining product selectivity. In Fischer–Tropsch (FT) synthesis, product selectivity is determined by the balance between both hydrogenation and C-C coupling abilities on metal surfaces. Typically, a catalyst with an excessively high $H_2$ dissociation or hydrogenation capacity may result in the creation of $CH_4$ with higher selectivity. Ni can be used, for instance, in the dissociation of CO. As stated earlier, the main active metals investigated for FT synthesis are Fe, Co, and Ru since they are appropriate for the synthesis of significant cumulative hydrocarbons. Ru is the most effective catalyst among these three metals for CO hydrogenation, and CO hydrogenation on Ru can take place at low temperatures. Ru has a higher affinity for long-chain hydrocarbons and a low affinity for $CH_4$; however, due to its high cost and restricted supply, Ru cannot be used as the primary active ingredient in large-scale industrial applications. Fe and Co have so far only been used in industrial FT processes as active metals. Ru is appropriate for basic study, nevertheless, as an excellent FT active metal that can offer significant insight into the catalytic reaction mechanism. Additionally, Ru is frequently employed in FT catalysts as a stimulant to speed up the reduction of Co or Fe because it is difficult to reduce Co or Fe precursors under normal circumstances [11]. Although Fe is less expensive than Co, Co catalysts are typically more active and selective to linear long-chain hydrocarbons. Co catalysts are also frequently more water-resistant than other types of catalysts. As a result, Co catalysts have drawn a lot of interest for the production of long-chain linear hydrocarbons from syngas, such as wax and diesel fuel. In contrast, Co catalysts can only function effectively at restricted temperature ranges and $H_2/CO$ ratios, whereas Fe-based catalysts can operate under a variety of temperatures and $H_2/CO$ proportions without significantly increasing $CH_4$ preference. It is important to note that, relative to Co- or Ru-based catalysts, Fe-based catalysts demonstrate vastly greater activity for the water–gas system process. This is beneficial for converting syngas made from coal or biomass that has a lower $H_2/CO$ ratio, but it is not desired for converting $H_2$-rich syngas made from natural gas or shale gas. For Fe catalysts to provide acceptable activity and selectivity, more substantial changes are typically needed, and quick catalytic deactivation is a major difficulty for Fe catalysts.

Solid oxygen carriers, which can be either single metal oxides or combinations of metal oxides, can be used to provide oxygen for this partial oxidation process. Zhu Xing et al. demonstrated the synthesis of syngas and hydrogen through two-step steam reforming of methane on a $CeO_2$-$Fe_2O_3$ oxygen carrier [12]. In their work, in a fixed-bed reactor, temperature-programed interactions between methane and the oxygen carrier $CeO_2$-$Fe_2O_3$ were carried out to find the ideal temperature for the gas–solid reaction that produces syngas. They noticed that the smooth onset of $CH_4$ conversion starts at about 500 °C using temperature as a function of reaction. However, below 650 °C, the conversion is really low. The fact that conversion increases quickly with a temperature over 650 °C suggests that reaction temperature has a significant impact on methane oxidation by solid oxides and that a higher temperature is required to produce syngas through a gas–solid reaction. This might be attributed to the fact that the rate of lattice oxygen migration from the bulk to the surface accelerates with increasing temperature. On the $CH_4$ conversion curve, they also noticed a slight peak at about 600 °C. $CeO_2$-$Fe_2O$ has a higher $CH_4$ conversion when the temperature exceeds 800 °C. Since oxidation reactions at low temperatures are typically aided by the surface adsorption of oxygen on oxides, this is expected to be caused by the well-released bulk lattice oxygen in $CeO_2$-$Fe_2O_3$. High-pressure experiments were carried out for the reduction and oxidation of oxygen carrier particles in a dedicated thermogravimetric analyzer as part of a study on Fe-oxide-based oxygen carriers for syngas production from methane by Niranjani Deshpande et al. [13]. The partial oxidation of $CH_4$ for syngas production using a Fe-oxide-based system is shown in Figure 1.

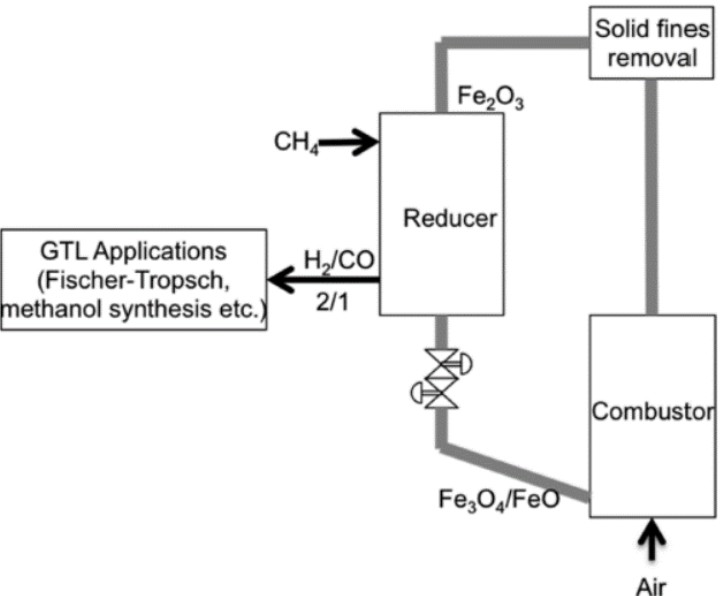

**Figure 1.** Illustration of the $CH_4$ partial oxidation-based Fe-oxide system for producing syngas. Reprinted from [13]. Copyright (2015), with permission from the American Chemical Society.

Due to their low cost and high magnetization capability, transition metals have generated interest [14]. Nevertheless, when contrasted to their respective mono-metal counterparts, bimetallic alloys have more applications and qualities, which have made them particularly attractive to researchers. Recently, there have been various attempts to use bimetallic catalysts in catalyzed reactions [15]. Bimetallic catalysts are considered successful because they are made up of two different metals with large dispersion and active sites due to the combination of their parent metals. Additionally, the creation of a solid solution improves the physical and chemical characteristics of bimetallic catalysts. A bimetallic catalyst exhibits a greater carbon yield than a monometallic catalyst, according to Pudukudy et al. [16]. Hydrothermal, pyrolytic, Sol–gel, sonochemical, radiolytic, microwave combustion, impregnation, microemulsion, and precipitation methods are a few of the physical and chemical processes that have been used to create bimetallic alloys. Researchers are still drawn to studying the synthesis of bimetallic magnetic alloys for the synthesis of syngas production [17]. The most prominent alloys used for syngas production comprise elements such as Ni, Co, and Fe. As catalysts, alloys perform better. Siddhartha Sengupta et al. synthesized $(Ni/Al_2O_3)$, $(Ni-Co/Al_2O_3)$, and $(Co/Al_2O_3)$ catalysts with 15 % metal content. They observed that the initial turnover frequencies of reforming $CH_4$ for the $Ni-Co/Al_2O_3$ catalysts were higher than those for $Ni/Al_2O_3$, indicating that the Ni-Co alloy sites are more active than the Ni sites [18]. Through calcination and reduction of hydrotalcite-like compounds comprising $Ni^{2+}$, $Cu^{2+}$, $Mg^{2+}$, and $Al^{3+}$, Dalin Li et al. created Ni-Cu/Mg/Al bimetallic catalysts that were then tested for the steam reforming of tar developed by a low-temperature pyrolysis of biomass. Even at a low temperature of 823 K, the catalyst provided almost an complete conversion of the tar [19]. The higher metal dispersion, greater number of surface active sites, better oxygen affinity, and surface alteration brought on by the production of small Ni-Cu alloy particles all contributed to this high performance. Non-supported alloys with their distinct morphological features were synthesized by Buthainah Ali et al. and used in catalytic biogas decomposition to produce syngas and carbon bio-nanofilaments at a reaction temperature of 700 °C and 100 mL $min^{-1}$. The behavior of the catalysts, $CH_4$ and $CO_2$ conversion, and the carbon generated during the reaction were studied [17]. Irrespective of metals, alloys, and oxides, magnetic materials are selected as the better candidates for the catalyst process due the above discussed reasons, along with their magnetic characteristics. Various magnetic nanomaterials and their composites could be utilized as catalysts for syngas production/conversion. The

next section will discuss the different synthesis methods that could be utilized for the preparation of magnetic nanocatalysts.

### 3. Synthesis of Magnetic Nanomaterials

To date, a variety of methods have been developed for the synthesis of magnetic nanomaterials and nanocomposites, viz. wet chemical route, microfluidic process, hydrothermal process, arc plasma discharge, chemical vapor deposition, microemulsion method, biogenic route, etc. Recently, there has been a heightened interest in synthesizing magnetic nanoparticles of uniform size for various technological applications [20]. Likewise, magnetic nanoparticle synthesis is one of the most critical challenges in tailoring particle size, shape, and crystalline structure. A wide range of magnetic nanostructures, including metals, metal alloys, oxides, and composite structures, have been synthesized using numerous techniques. Magnetic nanoparticles are prepared using three different methods, including (i) incipient wetness impregnation, (ii) precipitation, and (iii) modified sol–gel, to obtain different sizes of magnetic nanoparticles through synthesis. However, synthesis methodology plays a key role in controlling morphology, catalyst particle size, size distribution, and surface area. For example, A. Alayat et al. reported that a Fe/NS-I catalyst that was prepared using wetness impregnation and activated by CO has the highest activity and that its selectivity falls within the range of gasoline C6–C14 [21]. By contrast, the other Fe/NS catalysts show poor catalytic activity and selectivity when these Fe/NS catalysts were prepared using precipitation and modified sol–gel methods. The optimal preparation and activation methods to reach the highest catalytic activity and selectivity toward light hydrocarbons are with the Fe/NS-I catalyst activated by CO. In addition, the Fe/NS-I catalysts' selectivity favors aromatics in the C6–C14 range. In this section, we will describe some methods that provide excellent size and shape control for preparing magnetic nanomaterials, as shown in Figure 2.

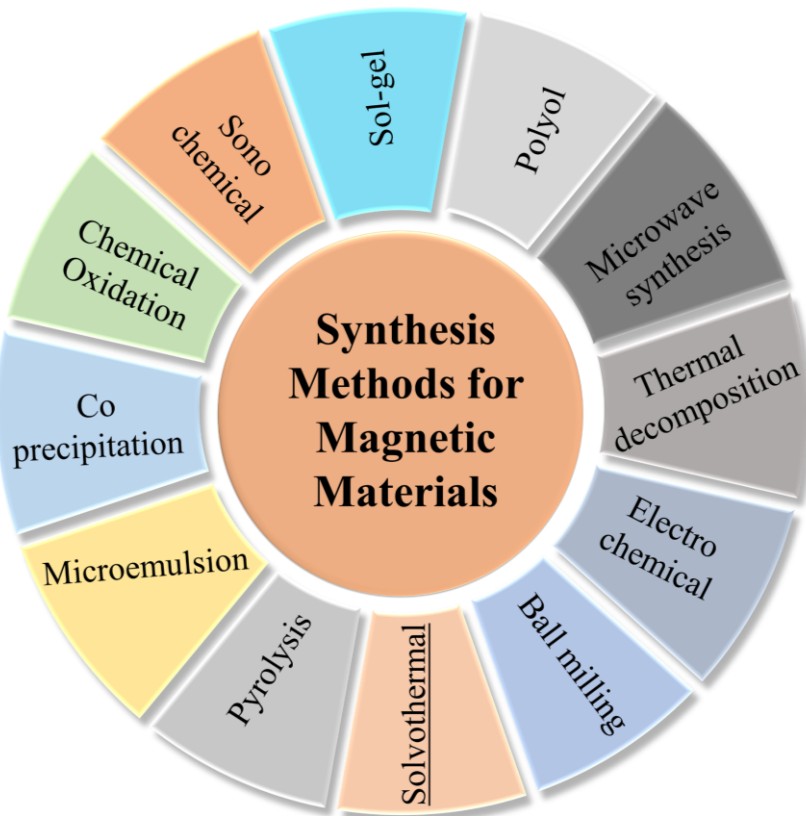

**Figure 2.** Different approaches for the preparation of magnetic nanoparticles.

Wet chemistry methods, which employ macro-scale equipment to carry out the reactions, enable batch production of nanoparticles. In comparison to microfluidic and biogenic synthesis, conventional MNP synthesis uses more energy [22]. Furthermore, it is more difficult to control morphology. Wet chemical routes are divided into hydraulic and non-hydraulic techniques, as shown in Figure 3, each of which has advantages and disadvantages depending on various variables, such as reaction temperature, pH value, precursor type, ratio of precursors, and nature of the base, which are the critical parameters that influence the size of magnetic nanoparticles [23]. With increasing pH, primary magnetite nanoparticles repel each other, thus becoming smaller [24]. The stoichiometry of the final magnetic nanoparticles could be altered by adjusting the ratio of precursors. The size of magnetic nanoparticles could be controlled by introducing a surfactant or by adjusting the reaction temperature. Wet chemistry method is considered one of the low-cost and simple chemical processes as it does not require any higher-cost equipment and processing. R. A. Frimpong et al. reported that the wet coprecipitation method used to prepare iron oxide nanoparticles gives a wide size distribution of magnetic nanoparticles [25] due to their chemical stability and biocompatibility compared to other metallic magnetic properties. Therefore, many investigations have focused on the development of large-scale production methods of uniform magnetic nanoparticles.

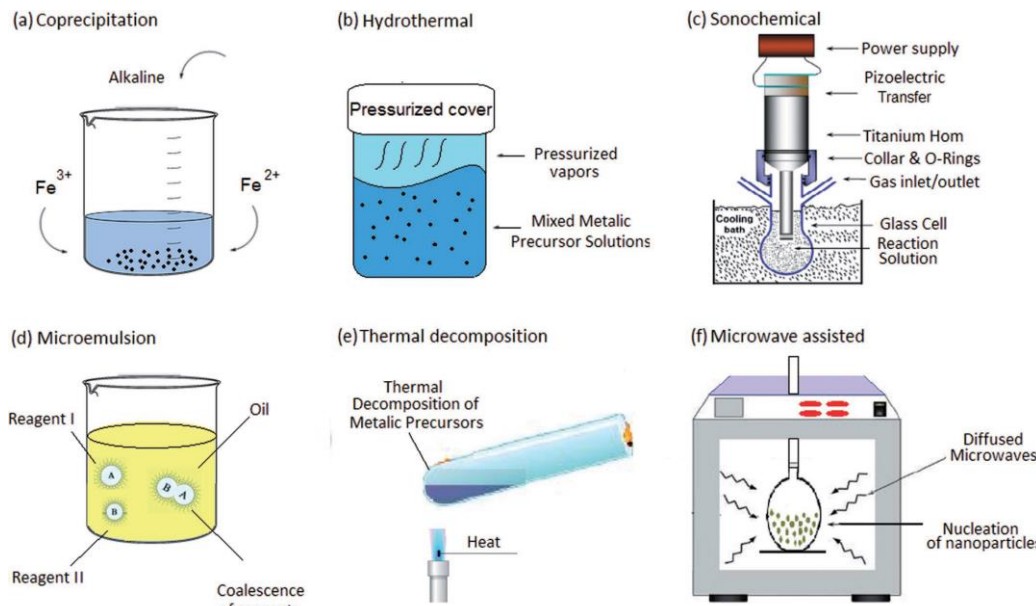

**Figure 3.** Schematics of distinct wet-chemistry hydrolytic and non-hydrolytic strategies for the synthesis of magnetic nanoparticles, (**a**) co-precipitation, (**b**) hydrothermal, (**c**) sonochemical, (**d**) microemulsion, (**e**) thermal decomposition, and (**f**) microwave assisted process. Reprinted with permission from [23], John Wiley & Sons, Inc.

In arc plasma technology, the plasma field provides the energy and reaction environment needed for atomic excitation, collision bonding, and more rapid formation of nanophase structures [26]. Using the arc plasma technique, high-temperature (>3000 degrees Celsius) plasma is generated between closely spaced electrodes under an inert atmosphere of helium or argon using a direct current. In this process, an electrode substance is evaporated by controlling the arc discharge, and the corresponding products are obtained upon cooling down on a cathode or a vacuum chamber wall [27]. Wang et al. reported that a minimum RL of $-56.3$ dB at 12.1 GHz and an absorption bandwidth of 5.2 GHz are obtained in the multi-step synthesis process [26]. Arc plasma is first used to prepare magnetic metal particles, which are then combined with carbon compounds. Methane ($CH_4$) is employed as a gaseous source of carbon in the reactor vessel.

Hydrothermal process is another easy and one-step way to develop hydrophilic magnetite nanomaterials using various precursors. There are many methods that have been used to develop magnetic metal oxide nanoparticles [28]. Hydrothermal synthesis of $Fe_3O_4$ nanoparticles in organic solvents is an advantageous process because of the homogeneity of the procedure and its comparatively low environmental impact. In a study by Misuthani et al., an aqueous solution containing ferrous and ferric ions is used as a starting solution, followed by precipitations of ferrous hydroxide ($Fe_3(OH)_2$) and goethite (a-FeOOH), which are precipitated in an alkaline solution as precursors [29]. The research by Zhang et al. demonstrated parameters that cause aqueous solutions to reach a critical or supercritical state, and these parameters are of interest because they allow the almost simultaneous reduction of two metal salts at a high temperature and high pressure [30]. They used the hydrothermal synthesis route to create magnetic CoPt alloy nanowires based on this method. Komarneni et al. first reported the synthesis of crystalline unit oxides, such as $TiO_2$, $ZrO_2$, and $Fe_2O_3$, as well as binary oxides, such as $KNbO_3$ and $BaTiO_3$, using the microwave hydrothermal method. Using microwave-hydrothermal conditions, the influences of several factors, such as chemical concentration, duration, and temperature, on crystallization kinetics at a microwave frequency of 2.45 GHz were investigated [31]. The reaction parameters, such as reaction temperature, reaction time, ratio of the precursors, and fraction of the surfactant, could be varied to obtain magnetic nanoparticles with different morphologies.

Solvothermal crystallization is one of the most consistent processes for crystal growth, and the resulting grains of magnetite have a significantly higher crystallinity than those made by other processes. The solvothermal fabrication process is similar to hydrothermal fabrication, which uses non-aqueous organic solvents. In solvothermal techniques, organic solvents, such as methanol, toluene, 1,4-butanediol, and amines, are frequently employed. Organic solvents are used as the solvents in these solvothermal processes. The solvothermal method's ability to control water-sensitive precursors is its most significant advantage. J. Li et al. simultaneously demonstrate that solvothermal techniques may be utilized in conjunction with microwaves and magnetic fields for the semicontinuous synthesis of materials with much increased repeatability and excellent quality [32].

The thermal decomposition process is another conventional process for the synthesis of various magnetic nanoparticles. However, the major disadvantage of this process is that it may require high-cost chemicals to produce magnetic nanoparticles, such as FePt, CoPt, and other related magnetic nanoparticles. A. G. Roca et al. evaluated the synthesis of monodispersed $Fe_3O_4$ nanoparticles by focusing solely on the thermal decomposition of organometallic precursors in organic solvents at high temperatures in the presence of surfactants. They yielded nanoparticles that are not hydrophilic in nature and cannot be dispersed in water, impeding their biomedical applications [33]. Other than the above discussed methods, a few more specifically designed synthesis methods for magnetic nanomaterials have also been successfully demonstrated, namely the sol–gel auto-combustion method, polyol process, chemical oxidation methods, and different deposition processes. However, each method and process has its own advantages and disadvantages. Depending on the requirements of the targeted particle's physicochemical characteristics, one can choose an appropriate synthesis process for the development of magnetic nanomaterials. Furthermore, experimental parameters, such as reaction temperature, precursors, precursors ratio, oxidation or reduction agent fraction, surfactant or stabilizing agent, reaction time, and post-thermal treatment, could be used to alter the physicochemical characteristics of magnetic nanomaterials.

## 4. Syngas Production and Conversion

Syngas production and conversion have been the focus of recent research. Several processes have been demonstrated for the production and conversion of syngas. Syngas can be created by gasifying waste, biomass, or coal in a high-temperature atmosphere. Syngas is a gaseous fuel created when a feedstock is partially oxidized in controlled

operating conditions with a deficiency in oxygen. In a gasifier, gasification generally consists of five steps. In order to produce an incredibly clean gas at the exhaust, drying must first take place below 100 °C, with the goal of lowering the moisture content of the feedstock. The dry matter is scorched to a temperature of about 240 °C during pyrolysis as a result of the vaporization of heavy volatile chemicals (tars). In addition to hydrogen and carbon monoxide, tars contain more sophisticated compounds. A solid carbonaceous substance known as charcoal, which is composed of fixed carbon-to-carbon chains, is also produced by this process along with tar gasses. The conversion of biomass into a more useful energy form can be achieved using three main processes: (i) gasification, (ii) pyrolysis, and (iii) combustion [34]. Similar solid fuel feedstocks, such as coal, biomass, and wastes, are used in the production of syngas using traditional gasification techniques. Additionally, non-gasification technologies that employ a single reforming process or a series of catalyst-assisted reforming processes can convert liquid and gaseous feedstocks into syngas. Such procedures are anticipated to be effective in in situ syngas production techniques, as described in this section. Using chemical looping gasification in biomasses, the generation of syngas is depicted in Figure 4a. Firstly, the biomass is processed and devolatilized as it enters the fuel reactor. Additionally, there are numerous simultaneous interactions between the various pyrolysis products, the gasification agent (steam or $CO_2$), and the oxygen carrier. When tar is gasified with steam, it generates syngas, CO, or $CO_2$, as appropriate. Thirdly, during the pyrolysis and gasification of the biomass, redox reactions take place between both the active phases of the oxygen carrier ($M_xO_y$) and the gases that are produced. The oxygen carrier plays a vital role in the production of syngas. The oxidization of the reduced oxygen carrier and the partial combustion of the biomass yield the heat required for gasification progressions. Moreover, the oxygen carrier possesses catalytic qualities that raise syngas quality and lower tar concentration. The main material in syngas generation is an oxygen carrier containing metal oxides, which is in charge of moving heat and oxygen from the air reactor to the fuel or gasification reactor, thereby avoiding the expense of pure oxygen or significant volumes of steam [35] Higher $CO_2$ conversion and lower carbon formation are the results of the reaction between the surface oxygen species and the surface carbon species, which can adsorb $CO_2$ molecules to create bidentate carbonate species. Fuel reduction technologies based on metal oxides, such as thermal carbon reduction, metal oxide-based selective oxidation reactions, and chemical looping processes, have been demonstrated. Chemical looping reform is a method for producing syngas that partially oxidizes methane by utilizing lattice oxygen [35]. The syngas production mechanism using metal oxides is shown in Equation (1).

$$M_xO_y + \delta\, CH_4 = M_xO_{(y-x)} + \delta\, (2H_2 + CO) \tag{1}$$

Gasification is a fundamental process for converting a feedstock into synthesis gas. The process of gasification is a thermochemical conversion that uses partial oxidation of the raw material to create a gaseous mixture that is primarily composed of hydrogen and carbon monoxide [36]. There are four stages to the gasification process, depending on the type of gasifier (drying, pyrolysis, oxidation, and reduction). In Figure 4b, a graphical diagram for different Fe/Ca ratios used to synthesize $Fe_2O_3/CaO$ with $H_2$-enriched syngas conversion [37].

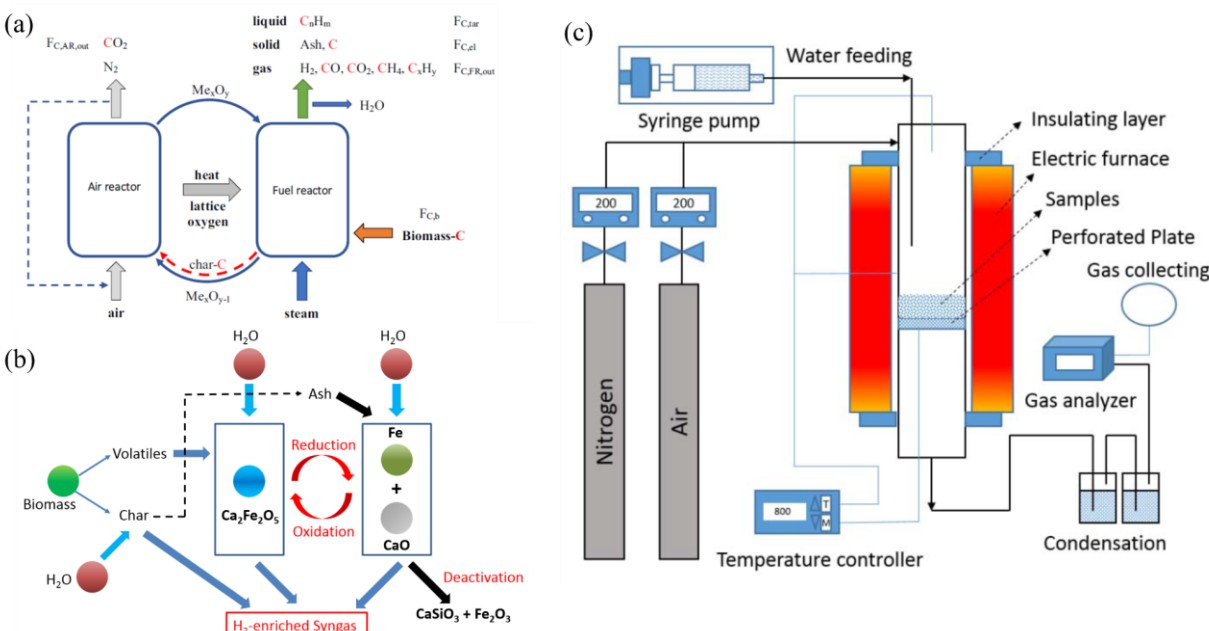

**Figure 4.** (**a**) The chemical looping gasification process. Reprinted from [35]. Copyright (2021), with permission from Elsevier. (**b**) Graphical diagram for different Fe/Ca ratios used to synthesize $Fe_2O_3$/CaO with $H_2$-enriched syngas. (**c**) Schematic diagram for different ratios of Fe/Ca used to synthesize $Fe_2O_3$/CaO with $H_2$-enriched syngas. Reprinted from [37]. Copyright (2020), with permission from Elsevier.

Figure 4c depicts the chemical looping gasification experimental technique, which consists of gas/water feeding, gasification reaction, condensation, syngas collection, and analysis. An electric furnace assembly with a stainless-steel tube reactor (inner diameter of 27 mm and height of 650 mm) was used as the gasification system in this experiment. Initially, the furnace was heated to a high temperature (600–900 °C). In the furnace, the stainless-steel reactor containing rice straw and OC was connected to a nitrogen flow (99.99%, 200 mL/min, and STP). A precise syringe pump was used to inject water into the reactor, and after the reactor was set up in the furnace, steam began to be produced and enter the reactor. At each run, 1 g of dried rice straw was mixed with 1 g of OC by mechanical stirring and reacted at the desired temperature for 30 min with a steam feeding rate of 0.1 g/min. The total steam was, therefore, affordable for the gasification of rice straw and the potential oxidation of reduced OCs, increasing the overall conversion efficiency. After the reaction, the solid residue was cooled down under nitrogen to room temperature and kept for further analysis. Based on this research, Quiang et al.'s study examined and compared the oxygen carrier durability, solid structure development, Fe/Ca ratio, temperature, and oxygen carrier cycling capacity to the pyrolysis and gasification processes. The trace extraction method resulted in the formation of two different kinds of calcium ferrites ($Ca_2Fe_2O_5$ and $CaFe_2O$) with various Fe/Ca ratios, with Fe and Ca being distributed equally throughout the oxygen carriers. When the Fe: Ca ratio was 1:1, the formed oxygen carrier ($Ca_2Fe_2O_5$) gave the highest hydrogen yield (23.07 mmol/g biomass) at 800 °C, benefiting from the one-step reduction and oxidation properties of $Ca_2Fe_2O_5$. A temperature of at least 800 °C was required for the complete redox reduction of $Ca_2Fe_2O_5$ during chemical vapor gasification. As a result, the high hydrogen selectivity of Fe: Ca = 1:1 ($Ca_2Fe_2O_5$) makes it a suitable oxygen carrier candidate, but the cycling stability should be improved to make biomass chemical chain gasification conversion a better application for the production of syngas [37]. Similarly, recent research by Liu et al. examined the CLG of microalgae with calcium ferrite or modified calcium ferrite in terms of syngas production and characteristics of OCs. Their findings demonstrated that $Ca_2Fe_2O_5$

was an appropriate material for microalgae because of its high selectivity for the production of synthesis gas [38].

Combustion is a chemical reaction between a fuel and an oxidant that makes an oxidized product. Generally, it is a reaction between hydrocarbons and oxygen to generate carbon dioxide, water, and heat. According to Zhang et al., the partial oxidation of fuel by chemical looping combustion can result in the formation of syngas [30]. $CaFe_2O_4$ and $Ca_2Fe_2O_5$ were discovered to be the two carriers that produced partial oxidation of solid fuels, the best among the four. The investigation on charcoal, however, may indicate a limitation with the respective carriers when using raw biomass because of the presence of various minerals, depending on its source. Besides, numerous investigations have also demonstrated the successful production of syngas using chemical looping gasification technology [39]. Likewise, Luo et al. obtained pure $Fe_2O_3/MgA_{12}O_4$ and iron ore with a temperature limit of less than 950 °C while using methane as a fuel [40].

Figure 5 illustrates the concept of indirect gasification technology with a combustion process [41]. In syngas production, a fuel mixture consisting mainly of hydrogen and carbon monoxide is burned, proving to be a promising fuel for combustion technology. According to Amin Paykani et al., the choice of base fuel, its physicochemical properties, and its physical conditions are all important factors that influence the synthesis of syngas produced by combustion systems [42]. M. Fiore et al. demonstrated that gasification is the only exothermic process and its heat release is necessary for water and carbon dioxide, as the products of several oxidation reactions are formed during this stage [43]. During cracking, carbonaceous particles become simpler molecules once they have been exposed to heat. To generate a gas compatible with an internal combustion engine, tars must first be broken down chemically, Otherwise, they could condense into sticky tars, which may cause damage to the valves. The reduction process, which assures the synthesis of hydrogen and carbon monoxide from water and carbon dioxide, is the last step. In order to achieve a high fraction of fuel gas at the exit, reduction is accomplished by passing the combustion products over a bed of extremely hot charcoal.

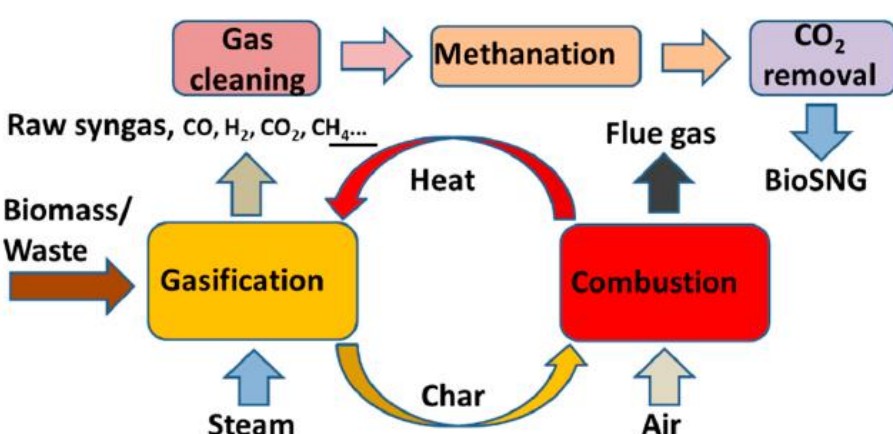

**Figure 5.** The concept of indirect gasification technology with the combustion process, reproduced from [41], with permission from MDPI Publisher.

Another process utilized for this purpose is pyrolysis, in which organic and plastic materials are thermally decomposed. During pyrolysis, materials undergo continuous physical and chemical changes. A wide range of applications is possible with pyrolysis technology. The chemical industry, for example, produces methanol, charcoal, and activated carbon by using this method. In addition to these, the pyrolysis process has several advantages over other energy recovery methods, such as the ability to use stone, ceramics, glass, or soil obtained during waste sorting. With this technology, a wide variety of feedstocks can be utilized. Consequently, it reduces landfill waste, landfill gas emissions, and the risk of water contamination. Moreover, building a pyrolysis plant is relatively simple [44]. Depending on the conditions associated with the process, pyrolysis can be classified into

three types: slow, fast, and flash pyrolysis. A variety of thermochemical and biological processes have been used to transform biomass into products with added value. Among those processes, pyrolysis is more convenient since it has several advantages in terms of storage, transportation, and flexibility in the solicitation. This includes engines, boilers, combustion appliances, turbines, etc. Generally, it can be divided into components that generate only heat and biochar (by using slow pyrolysis), and units that produce both biochar and bio-oils (using fast pyrolysis). Additionally, flash pyrolysis is a quick pyrolysis unit that transforms biomass into bio-oil [43,44]. Typical pyrolysis systems comprise a pyrolysis reactor, an extraction unit, and equipment for pre-processing lignocellulosic residues.

## 5. Tuning of Experimental Parameters

When developing nanocatalysts, which are employed in numerous chemical reactions to speed up the reaction and improve product efficiency, nanomaterials are frequently used because they have special catalytic capabilities. The effects of experimental factors, such as nanoparticle size, shape, distribution, and preparational methods, on their catalytic properties have been investigated by various researchers [45]. Basically, research has been conducted to comprehend the mechanism of action of nanocatalysts or the activity of catalysts, as well as the selectivity of nanocatalysts. Due to their distinctive features, nanoparticles have attracted attention as a material for advanced biofuel processes. Nevertheless, the use of nanoparticles in the production of bioethanol is still in its infancy. The morphology, physicochemical properties, and durability of nanoparticles are significantly impacted by a number of parameters throughout the synthesis process. Therefore, operational factors have a great impact on the synthesis process. The performance of nanoparticles is influenced by a variety of factors during the production of biofuel. The synthesis method, pressure, temperature, medium pH, nanoparticle size, and other variables are crucial [46].

### 5.1. Temperature

Since temperature affects the geometry of nanoparticles, comprising their size, shape, and endurance, temperature is a significant factor in the synthesis of biofuel. The metabolism and proliferation of cells are greatly influenced by temperature when using biological methods to produce syngas [47]. The solubility of CO, $H_2$, and $CO_2$ in the fluid phase is influenced by temperature. For different microorganisms, different temperatures are ideal for syngas production. Physical and chemical techniques require temperatures higher than 300 °C, but biological methods utilize temperatures that are low to moderate (<100 °C), or even ambient. Besides this, based on the synthesis technique, the calcination temperatures for different metallic nanomaterials vary between 100 and 700 °C. The effects of temperature influence the geometry of nanoparticles, the rate of reaction during the catalytic reaction, the durability of nanoparticles, and the structure and size of nanoparticles. Reaction rate and temperature are tightly correlated, suggesting that reaction rate is slow at a relatively low temperature and speeds up as the temperature increases. Compared to nanoparticles produced at reduced temperatures, such as 25 °C, those produced at higher temperatures, such as 100 °C, are more stable and develop to their proper size. A higher temperature can speed up the production process [48]. Transesterification is a synthesis process that is used to prepare straight oils from renewable sources for biodiesel production [43,45]. When methanol or ethanol are present, transesterification processes begin at temperatures between 50 and 80 °C with sodium hydroxide (NaOH) functioning as a base catalyst. The impacts of catalyst calcination temperature and catalyst bed temperature on the constitution and productivity of syngas were studied by Shuangxia Yang et al. [49]. Their studies focused on the synthesis of hydrogen-enriched gas from biomass in a two-stage fixed-bed reaction system using a Fe/CaO catalyst formed from layered double hydroxides as the precursor. At a modest temperature range of 600 °C, the Fe/CaO catalysts displayed uniform morphological characteristics and crystalline size, and they had outstanding physicochemical characteristics, such as $H_2$ reducibility and $CO_2$ absorption capacity. This contributed to

their high catalytic performance in terms of the production of syngas, which resulted in a syngas yield of 44.6%, an $H_2$ composition of 35.7%, and an $H_2$/CO ratio of 1.36 in the range. Although the $H_2$/CO molar ratio decreased to 0.99 as a result of additional $CO_2$ released from the decomposition of $CaCO_3$ at higher catalytic temperatures, the syngas yield and $H_2$ yield increased up to 63.0 weight percent and 172 mL/g of biomass, respectively, as a result of the augmented supplementary cracking and realigning reactions of volatiles in the existence of the Fe/CaO-600 catalyst. The gaseous concentration and gaseous yield of the syngas conversion of biomass pyrolysis in the presence of the Fe/CaO catalysts calcined at various temperatures are shown in Figure 6.

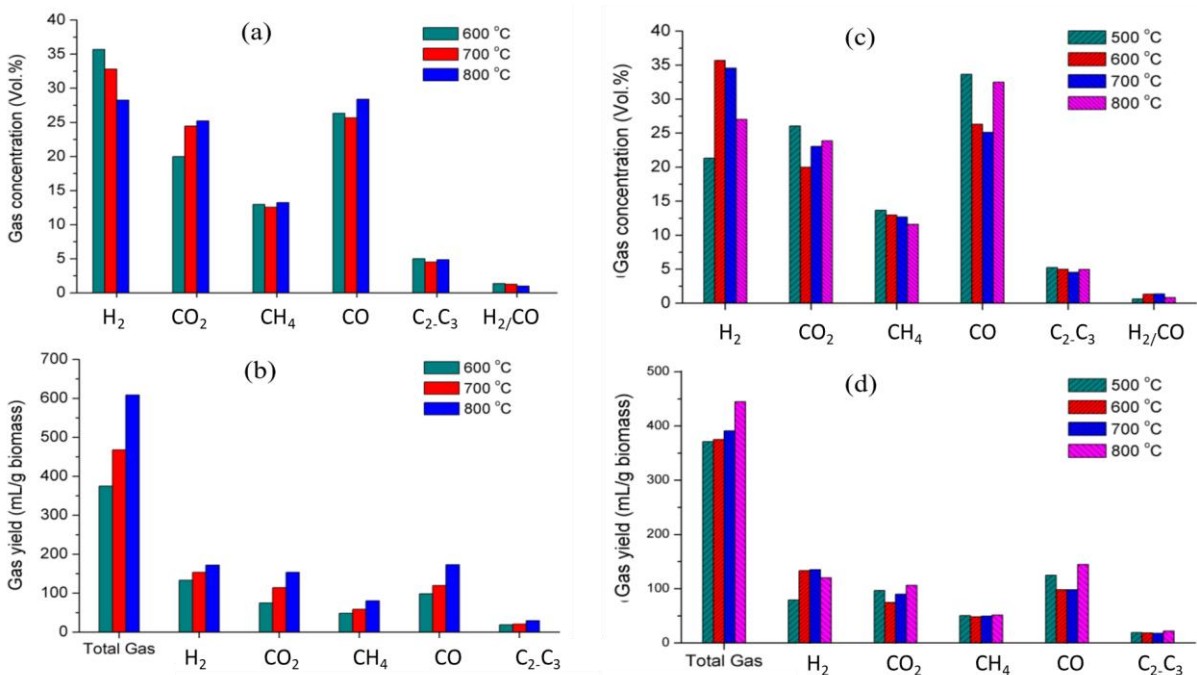

**Figure 6.** Bar diagram of (**a**) gaseous concentration, (**b**) gaseous yield of the syngas conversion of biomass pyrolysis, effect of catalytic bed temperature on (**c**) concentration, and (**d**) yield of the syngas. Reprinted from [49]. Copyright (2017), with permission from Elsevier.

*5.2. pH*

Due to the significant role that pH variation plays in the formation of nucleation centers, pH values have an effect on the shape, size, synthesizing pace, and homogeneity of particle dispersion of nanomaterials. Small nanoparticles are evident at a pH of 6, but when the pH rises from 7.0 to 11.0, the average nanoparticle size decreases [50]. The size and form of nanoparticles are influenced by the pH of the medium: a medium with a basic pH offers a quick growth rate, faster productivity, and enhanced reduction rates, and it regulates the size, synthesizing rate, homogeneity of particle distribution, and morphology of nanomaterials. The typical size of nanoparticles drops when the pH rises from 7.0 to 11.0, while microscopic nanoparticles are evident at a pH of 6. This not only impacts the regularity of particle distribution and synthesizing speed, but it also influences the growth of nucleation sites in the morphology of the particles [50–52]. Z. Fakhroueian et al. studied the impact of the pH factor on the production of nanocatalysts, nanocomposites, and nanotubes using the hydrothermal, coprecipitation, and sputtering techniques of $Cu_{O_x}$ $NiO_x$ for syngas processes in the petroleum sector [53]. According to their research, the pH factor can alter the original texture, surface characteristics, and material properties. For instance, 25 percent of NiCo/CeAl nanocomposites at a pH of 9 can yield nanotubes and nanocomposites with the finest adaptive modulation, 99 % $CH_4$ conversion, and 91 % $H_2$ selectivity in the syngas reactions; however, at a pH of 7, it obtains a simple nano-spherical assembly, containing 98% of $CH_4$ conversion and 93.40 % of $H_2$. The pH of the culture has

a considerable impact on aerobic metabolism, cell proliferation, and product distribution in biological processes. An effective approach to promoting the formation of syngas is a pH shift.

### 5.3. Catalyst Preparation Process

The coprecipitation method, microemulsion, thermal breakdown, hydrothermal synthesis, synthesis employing plant materials, synthesis incorporating biological organisms, and so forth are instances of nanoparticle synthesis procedures. Every method has a unique set of benefits and drawbacks [54]. Different synthesis methods have an impact on the efficiency of nanomaterials for the production of syngas. The most efficient processes for producing biodiesel are the ultrasonic reactor, batch reaction, lipase-catalyzed technique, and supercritical process. With these techniques, transesterification is used to synthetically create pure oils from renewable resources. Enzymatic hydrolysis, preprocessing, and fermentation are several steps required for the production of bioethanol. Various pretreatment techniques, including physical, chemical, physicochemical, and biological procedures, have been employed for the conversion of biomass. Different synthesis techniques play a crucial role in the performance of nanoparticles in the conversion of biomass [55]. The generation of biofuel and the effectiveness of particles are severely affected by their synthesis methods [56].

### 5.4. Pressure

In the formation of nanomaterials and biofuels, pressure is a significant factor. Typically, the reaction medium is subjected to the proper pressure in order to produce the desired geometry, size, and coalescence of nanoparticles. Particle sizes rise as a result of the high pressure. In order to regulate the levels of quick deployment and decrease, pressure can also be applied [55,57]. The effects of adding ferric chloride ($FeCl_3$) to biodiesel on the performance and emission parameters of diesel engines were studied by Kannan et al. [57]. The rate of heat release and cylinder pressure were reported to have increased. While the thermal efficiency of the brakes increased by 6.3%, it was also determined that the specific fuel consumption decreased by 8.6%.

## 6. Utilization of Magnetic Nanomaterials

Magnetic nanomaterials of Fe, Ni, Co, and various combinations with other active nanomaterials and magnetic nanoparticles are found to be better candidates for use as the catalysts for syngas production/conversion process through various methods. This section discusses the progress made on magnetic nanomaterials containing Fe, Ni, and Co as the catalysts for syngas production/conversion process.

### 6.1. Fe-Based Magnetic Materials

For the synthesis of syngas, iron (Fe) is a promising active-phase material due to its lower cost than noble metal catalysts, abundance, low toxicity, and high reliability in tar breaking [58]. Due to the presence of mono-Fe and rhodium in the catalysts, conventional Rh-Fe catalysts often show greater alkane or methanol selectivity, and as a result, their specificity to higher alcohols is poor. Tong Han et al. synthesized an alternative catalyst. $YRh_{0.5}Fe_{0.5}O_3$ with a perovskite structure was layered on $ZrO_2$ as a solution to the conflict [59]. The homogenous and highly dispersed Rh-Fe alloys were responsible for the remarkable specificity and productivity of $YRh_{0.5}Fe_{0.5}O_3/ZrO_2$. $YRh_{0.5}Fe_{0.5}O_3/ZrO_2$ demonstrated excellent stability as well. Thomas E. L. et al. examined the hydrogenation of $CO_2$ into methanol using Ni-Fe-Ga alloys to generate $Ni_2FeGa$, and they observed that $Ni_2FeGa$ performed the best out of all the evaluated catalysts in terms of methanol output. At low processing temperatures and pressures, the efficacy of the produced $Ni_2FeGa$ yielded performance similar to commercial products, i.e., $Cu/ZnO/Al_2O_3/MgO$ [60].

Mixing different transition-metal elements is an efficient technique to boost catalytic activity, together with nano-structuring/crystallization to raise the bulk activity. When

methane and solid oxides (oxygen carriers) combine, syngas is produced. The reduced oxygen carriers can then be replenished by a gaseous oxidant, such as water and air [61]. Chemical looping ignition is regarded as a very effective technique for capturing $CO_2$. Since $CO_2$ segregation and capture are inherent to the process, it also offers a far lower energy penalty and cheaper cost than alternative $CO_2$ capture devices. Combustion occurs in the chemical looping process without the fuel and air coming into contact. Metal oxide ($Me_xO_y$), which is converted to $Me_xO_{y-1}$, delivers the ignition oxygen to the fuel reactor. Air is used to replenish the metal oxide inside a network of connected air reactors. The fuel is oxidized inside the fuel reactor to $CO_2$ and $H_2O$ [62]. Ral Pérez-Vega investigated the ignition of syngas using a Mn/Fe-based oxygen carrier and demonstrated complete fuel transformation to $CO_2$ and $H_2O$ at fuel unit temperatures exceeding 930 °C. Moreover, the $CH_4$ ignition reached the maximum burning rate of 78% at 966 °C. In addition, they reported that the fuel reactor was heated to a temperature higher than 1000 °C in order to complete $CH_4$ combustion [63].

Jiao Zhao et al. synthesized a Fe-N-C non-precious electrocatalyst, and a flow rate of 24 mL $min^{-1}$ of very pure $CO_2$ was delivered into the cathode chamber for 30 min prior to electrolysis [5]. At the same time, a magneton was used to agitate the electrolyte in the cathode chamber. Additionally, they stated that, compared to a resin made of carbonized melamine formaldehyde, the non-precious Fe-N-C electrocatalyst demonstrates well-pleasing movement of CO and $H_2$ generation ($FE_{CO}$ = 74%, $FE_{H2}$ = 25%) at a lower overpotential of 0.6 V than others. Additionally, they reported that, compared to a resin made of carbonized melamine formaldehyde, the Fe-N-C non-precious electrocatalyst exhibits well-pleasing activity of CO and $H_2$ generation ($FE_{CO}$ = 74%, $FE_{H2}$ = 25%) at a subordinate overpotential of $-0.6$ V, demonstrating the better selectivity of $CO_2$ reduction reactions [5]. Mesostructured titania and silica were synthesized by Claudio Cara and colleagues as support materials for the purification of $H_2S$ from syngas. Fascinatingly, it was found that both ultrasmall aspects were extremely active, sensitive to $H_2S$, and regenerable. Under a syngas environment, the two composites' active phases differed noticeably, with the amorphous silica-based composite performing better when there were Fe infusions [64]. Aluminum magnesium spinel ($MgAl_2O_4$) was developed by G. V. Pankina et al., who employed Fe-K as the activator, and its particular surface areas were examined in relation to the physicochemical characteristics and dynamics of Fe-K [65]. They observed that syngas reduction occurs predominantly when the catalyst's magnetization is higher.

Gihoon Kwon et al. developed a Fe-impregnated bentonite, which was used as a catalyst in the $CO_2$-assisted pyrolysis of grass cut to increase the production of syngas, quality of bio-oil, and sorptive properties of biochar. Similar to the pyrolysis of freshly grass cut, the production of hydrogen in the pyrolysis of the biochar under the $N_2$ condition begins at 500 °C (Figure 7(I)a). Yet, compared to the single pyrolysis of grass cut, it creates more $H_2$ with a maximum intensity at 700 °C, which is >2.5 times higher. Figure 7(I)b shows the concentration of $CH_4$ for the biochar. From their results, it is evident that the existence of bentonite accelerates the formation of $H_2$, particularly between 630 and 700 °C. At the corresponding temperatures, bentonite is also found to have a similar impact on CO generation, as shown in Figure 7(I)c. Figure 7(I)d displays the outcomes of the gas investigations from the co-catalytic pyrolysis of the Fe-bentonite [66]. In the presence of $N_2$, the pyrolysis of the Fe-bentonite produces more $H_2$ than the pyrolysis of the grass cut and biochar, with two peaks at 500 and 700 °C (Figure 7(II)a). The authors concluded that the Fe-impregnated bentonite is effective in pyrolyzing biomass to increase the generation of syngas and improves the standard of pyrogenic products.

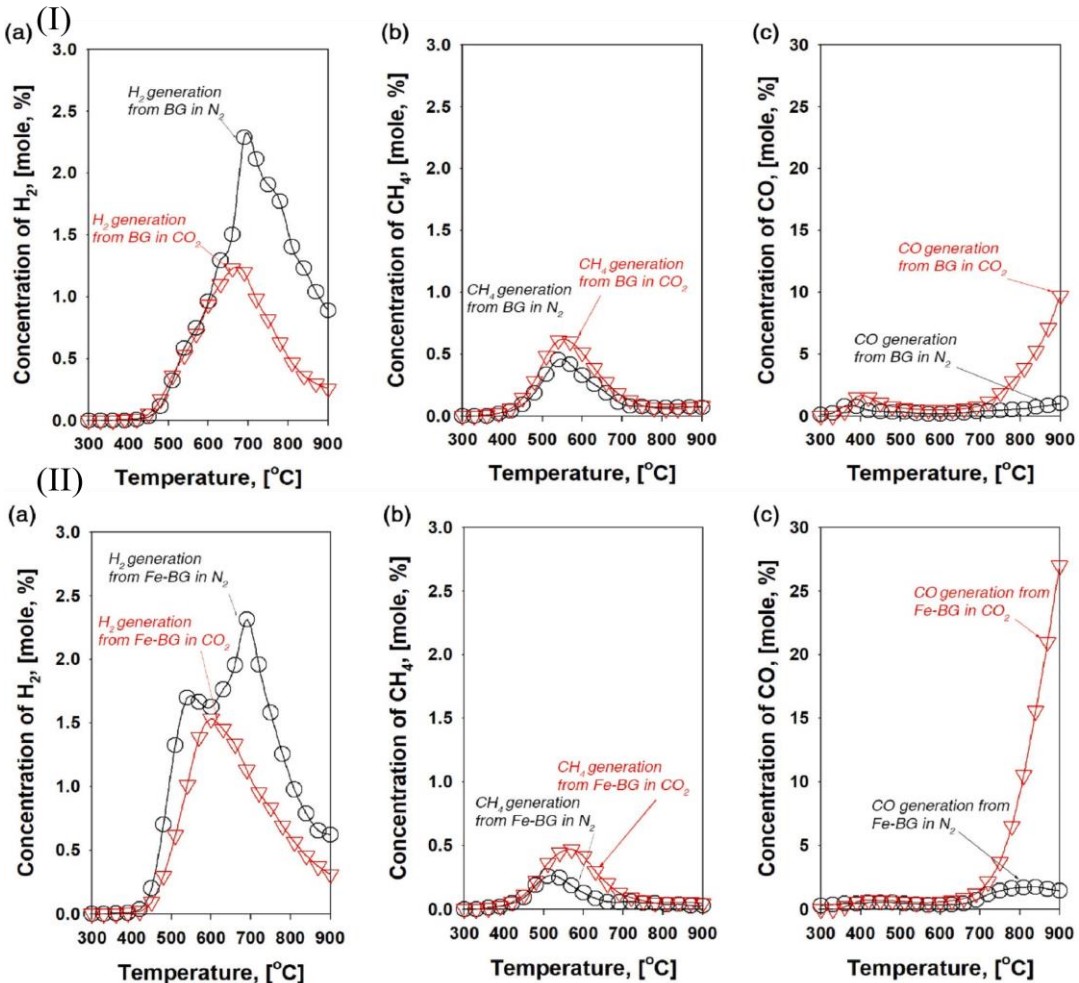

**Figure 7.** (**I**) Concentrations of $H_2$ (**a**), $CH_4$ (**b**), and CO (**c**) formed during the pyrolysis of grass cut under the $CO_2$ and $N_2$ conditions. (**II**) Concentrations of $H_2$ (**a**), $CH_4$ (**b**), and CO (**c**) formed during the pyrolysis of Fe-bentonite under the $N_2$ and $CO_2$ conditions. Reprinted from [66]. Copyright (2022), with permission from Elsevier.

### 6.2. Ni-Based Materials

The research on nickel (Ni) has focused on enhancing reforming catalysts' durability to manufacture syngas on a large scale [67]. To improve partial oxidation, hydrogenation, dry reforming, and steam reforming, Ni-based catalysts are frequently utilized. They are favored for the reforming process due to their great stability and catalytic activity, as well as the fact that they are less expensive than noble metal catalysts [68]. On account of coke formation on catalysts and Ni sintering, Ni catalysts are known to be susceptible to deactivation. Numerous research studies have proposed that improving catalyst stability might require changing the support, adding functional metals, and using the right catalyst preparation techniques [69]. Mohammad Yusuf et al. synthesized Ni- based alloy catalysts sustained on mixed oxide ($Al_2O_3$/MgO) with the inclusion of tungsten. They noticed that a 4% by weight of tungsten loading resulted in the best performance and that the Ni-W bimetallic alloy catalyst still produced above 90% conversion of greenhouse gasses after 12 h [70]. By studying the catalytic pyrolysis of a herbal residue using a Ni-doped Fe/Ca catalyst, Jin Deng et al. demonstrated that the strong interaction between Fe and Ca produced $Ca_2Fe_2O_5$, which prevented the formation of $CaCO_3$ and $CO_2$ and aided in the transition of tar and char. At 5% of the Fe-Ca catalyst, the $H_2$ output improved to 54.6 mL/g. Ni increased the dispersion of the catalyst and aided in the adsorption of alkaline ions while forming a stable $Fe_3Ni_2$ alloy with Ca. The yields of $H_2$ and CO improved from

54.6 mL/g to 63.4 mL/g for 0.5% of Fe/Ca and from 70.0 mL/g to 95.5 mL/g for 0.5% of Ni-Fe/Ca, respectively [71].

A multi-technique approach was used by L.B. Raberg et al. to investigate the potential relationship between support basicity, Ni/support activity, and stability of catalysts for the dry reforming of propane to synthesis gas. After being exposed to a magnetic field, ferrous metals retain a leftover magnetic moment at zero field; nonetheless, when the particle diameter falls below a particular size, the ferromagnetic particles transform into superparamagnetic particles [72]. The dry reformation of lignin with embedded Ni nanoparticles during microwave irradiation was prepared by M.V. Tsodikov et al., who adopted two distinct methods: impregnation and deposition [73]. In both instances, Ni particles with a size of 6 nm developed on the interface of the lignin. The dry reformation of lignin with microwave assistance was controlled by a variety of nanocatalysts. The first method's physical mixing of a carbon sorbent with a large dielectric loss factor and a specimen of lignin containing Ni was treated mechanically with a microwave, and the outcome was a 65-weight percent reformation of the lignin to synthesis gas with a $H_2$/O ratio of 1/1. The products contained 80–90 weight percent of syngas. It was observed that the microwave treatment significantly boosted syngas yield during reforming compared to convection heating. It was found that various deposition techniques produced X-ray amorphous Ni particles with noticeably varied magnetic characteristics. Superparamagnetic Ni0 particles developed by metal vapor synthesis showed the highest ability for absorption using the microwave technique, which was adequate for the plasma to be derived as well as for the reforming temperature to be attained without the need for an additional sorbent; this was in contrast to the completely paramagnetic sample made from a nickel acetate solution [73]. It is quite well known that highly distributed metals with magnetic characteristics have a large absorbency in microwave assistance.

In a two-step gasification of sewage sludge at a treatment temperature of 500, 700, and 900 °C, Chao Gai et al. showed catalyst selectivity toward the primary gas components [74]. They observed that the experiments using the hydrochar without Ni nanoparticles showed lower discernment (35–46%) of total gas composition toward $H_2$ and $CO_2$ production of 19–35%. The Ni nanoparticles synthesized with the hydrochar catalysts, on the other hand, all displayed better catalytic activity and selectivity for $H_2$ production. From their report, it is demonstrated that Ni nanoparticles are useful for gasifying biomass to produce $H_2$-enriched syngas. The authors postulate that the hydrochar-supported, evenly disseminated metallic Ni nanoparticles are responsible for the remarkable catalytic activity and selectivity toward $H_2$. Figure 8 shows the two-stage gasification of the sewage sludge's catalyst selectivity toward the major gas components at a treatment temperature of 500, 700, and 900 °C.

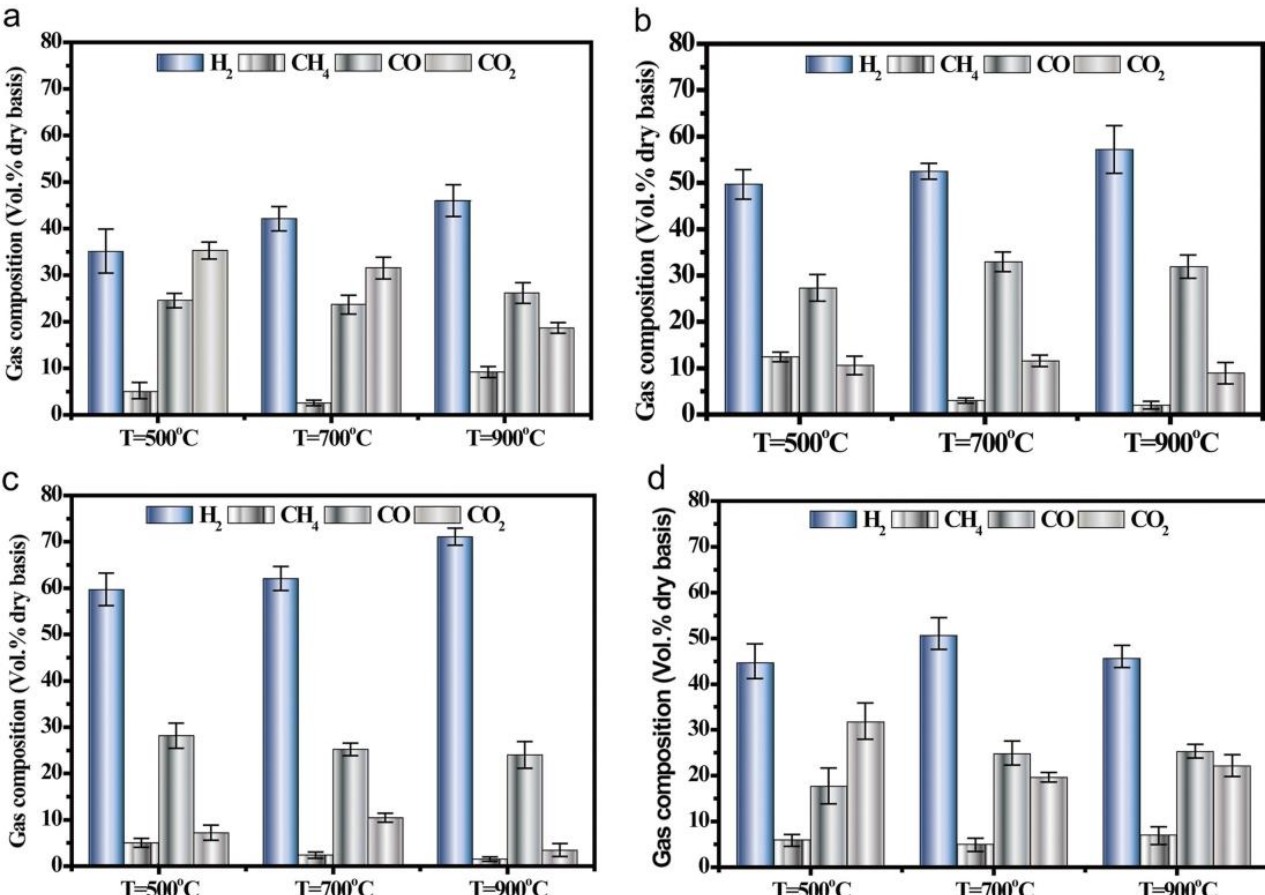

**Figure 8.** The gas product selectivity from the catalytic gasification of sewage sludge using catalysts from (**a**) hydrochar and (**b–d**) nickel@hydrochar with different concentrations of nickel (0.1, 0.5, and 1.0) at different gasification temperatures from 500 to 900 °C. Reproduced from [74]. Copyright (2019), with permission from Elsevier.

The core–shell LaMer model is largely acclaimed as being reliable, which is comprised of a hydrophobic core and a hydrophilic shell, and the model was applied by Chao Gai et al. for the synthesis of hydrochar-supported Ni nanoparticles. The interface of the hydrochar is hydrophilic with a dispersion of -OH and -C=O, such as carbonyl, hydroxyl/phenolic, and carboxylic functions. The carbonyl, hydroxyl/phenolic, and carboxylic functional groups on the surface of the hydrochar engage with $Ni^{2+}$ ions to absorb them when they are added to the hydrothermal liquid through ion exchange interactions [74]. Figure 9 displays a schematic illustration of the hydrochar-supported nickel nanoparticles.

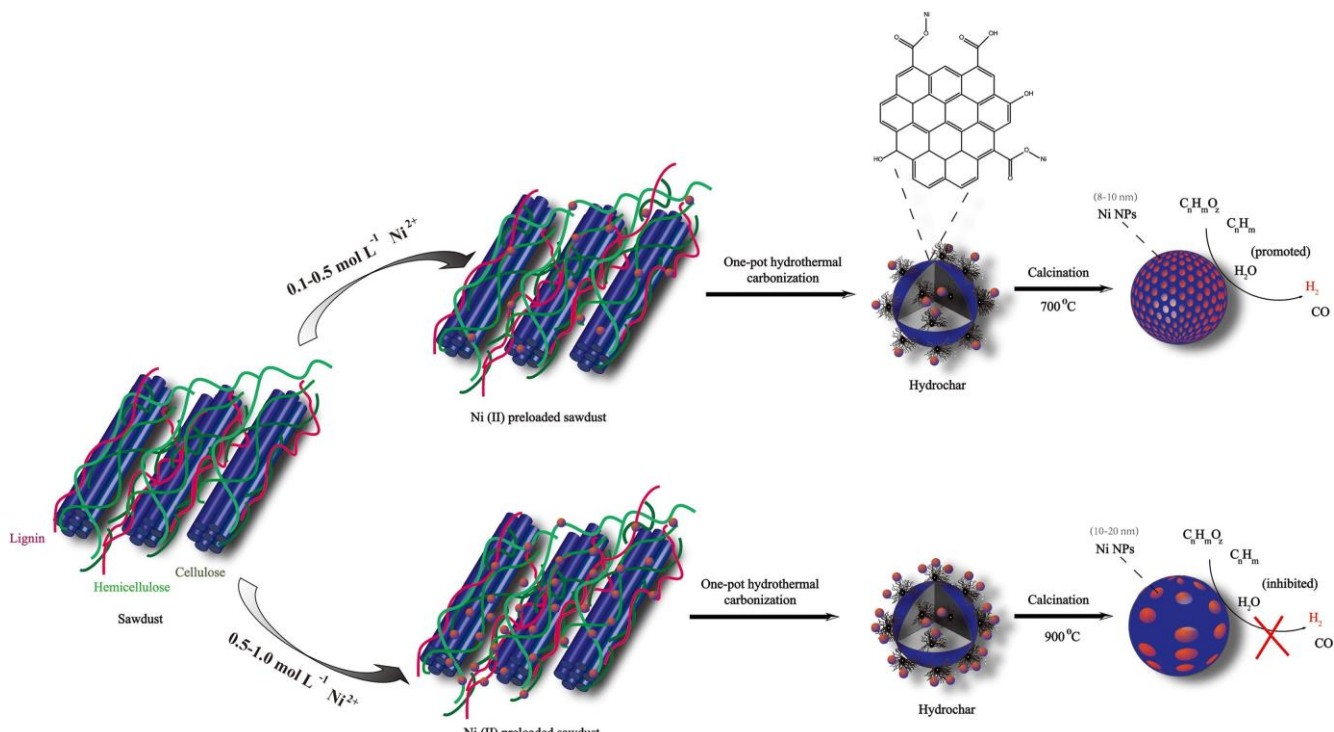

**Figure 9.** Schematic illustrations of hydrochar−supported nickel nanoparticles synthesized from a lignin-rich precursor biomass. Reproduced from [74]. Copyright (2019), with permission from Elsevier.

Ziwei Li et al. fabricated Ni@Ni−Mg Phyllosilicate core/shell catalysts by treating hydrothermally Ni@SiO$_2$ with Mg(NO$_3$)$_2$ [75]. The Ni@Ni-Mg Phyllosilicate core/shell catalyst treated for 10 h had the best catalytic performance out of all the Ni@Ni-Mg Phyllosilicate catalysts synthesized under different hydrothermal durations, with stable CO$_2$ and CH$_4$ conversions of around 81 and 78%, respectively. Its stability for the dry reforming of the methane reaction is shown in Figure 10(I). The most potent catalyst (treated for 10 h) was taken through testing for approximately 100 h, and it demonstrated conversions of CO$_2$ and CH$_4$ of 80 and 78%, respectively. The H$_2$/CO ratio was approximately 0.96. This demonstrated the stability of the catalyst, treated for 10 h for the dry reforming of the methane process, at 700 °C. The TGA analysis for the used catalysts after a 20 h process is shown in Figure 10(II). For the 20 h testing period, there is negligible weight loss for Ni@SiO$_2$. This may be due to the resistance influence of SiO$_2$ to Ni during sintering, which slows down the reaction of CH$_4$ breakdown. Due to the prevalence of CH$_4$ breakdown on sintered NiO, the catalyst used before calcination (befcal-NM-24 h) shows the most intense carbon deposition (26.1%). Despite being lower than that of the catalyst treated for 2.5 h (M-2.5 h) and 10 h (M-10 h), the weight loss for the after-calcination (aftcal-NM-24 h) catalyst is still higher at 6.3%. The M-2.5 h and M-10 h catalysts exhibit significantly less weight loss when compared to the befcal-NM-24 h catalysts, while having identical porosity and Ni exposures but higher basicity. Additionally, although having a weight loss of 10.4%, the catalyst treated for 24 h (M-24 h) loses 2.5 times less weight than the befcal-NM-24 h catalyst since it has a similar porosity and surface Ni exposures. These findings demonstrate the magnesium entity's potential for reducing the issue of carbon deposition, as illustrated in Figure 10(III). The authors performed a TEM analysis to examine the morphologies of the used catalysts. Following 20 h of reaction at 700 °C, cross-linking across silica shells emerges for both the Ni@SiO$_2$ and aftcal-NM-24 h catalysts, as can be seen in Figure 10(III)(a,b). This is mediated by the hydroxylation process between -(Si-O-Si)- assemblies on the silica shells, which results in the formation of three-dimensional systems when water is present. Due to the low H$_2$/CO ratio, this water is formed through a reverse water–gas shift reaction.

From Figure 10(III)(c), a substantial amount of carbon nanotubes can be seen. This might be a result of NiO phase sintering. For the catalysts treated for 2.5 h (M-2.5 h) and 10 h (M-10 h), there is no cross-linking within the silica shells, as shown in Figure 10(III)(d,e). In the core of these two catalysts, Ni is well diffused, with the outer layer consisting of a needle-like Ni@Ni-Mg Phyllosilicate phase. Moreover, for the catalysts treated for 2.5 h (M-2.5 h), a tough outer silica shell is still discernible, whilst for the catalysts treated for 10 h (M-10 h), the majority of the porous Ni@Ni-Mg Phyllosilicate species replace the silica shell of Ni@SiO$_2$. This makes it possible for the core portion to have great accessibility to active Ni while still limiting their sintering, which leads to strong $CO_2$ and $CH_4$ conversions and minimal carbon deposition. On the other hand, a significant number of carbon nanotubes and effective sintering of Ni /NiMg solution occur, as shown in Figure 10(III)(f), which lead to a lower catalytic activity and lower stability.

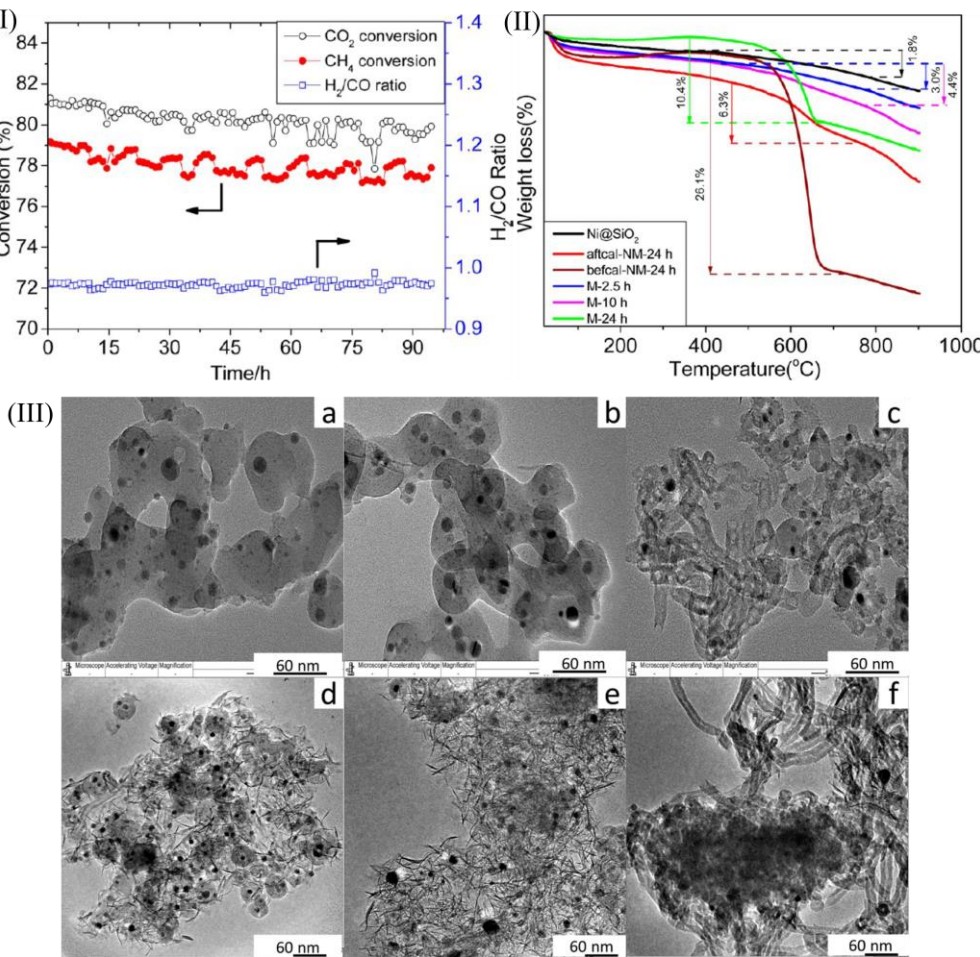

**Figure 10.** (**I**) Stability of the catalyst treated for 10 h and calcined at 700 °C. (**II**) Thermogravimetric analysis (TGA) for the utilized catalysts with various structures. (**III**) TEM images of the utilized catalysts after 20 h of reaction (**a**) Ni@SiO$_2$, (**b**) aftcal-NM-24 h, (**c**) befcal-NM-24 h, (**d**) M-2.5 h, (**e**) M-10 h, and (**f**) M-24 h. Reprinted with permission from [75]. Copyright (2014), with permission from the American Chemical Society.

### 6.3. Co-Based MATERIALS

According to a report, Co-based alloys are efficient and have a high hydrocarbon production rate, making them a suitable material for use in syngas conversion [76]. Considerable efforts have been made to develop a range of different metal alloy materials to reduce the total cost and promote the catalytic efficiency of catalysts. Co-Cu bimetallic catalysts are regarded as the most viable catalysts due to their high affinity to higher alcohol and cheap cost. Cu-decorated Co-based catalysts and Co-decorated Cu-based catalysts are the two

forms of Co-Cu bimetallic catalysts. Through theoretical and practical research, Prieto et al. demonstrated that a Co-Cu alloy phase having Co-rich ratios is the best catalyst interface for long-chain alcohols [77]. Using a steady-state isotopic transient kinetic analysis, the results suggest that the addition of Cu into Co significantly promotes the formation of more $C^{2+}$ oxygenates by blocking a significant fraction of Co sites for hydrocarbon synthesis [78]. Using radio-frequency magnetic fields, Hoang M. Nguyen et al. created binary (Co-Cu)- and ternary (Co-Cu-Ni)-based alloys. All prepared catalysts were stimulated under radio-frequency ignition to transform $CO_2$ and co-reactants, such as $H_2O$ and methane, into syngas ($H_2$ and CO) at a low temperature of 400 °C [79]. They also concluded that, in all the performance conditions, the Cu-Co sample had the best catalytic stability and performance. According to their research observations, the catalyst system's inclusion of a magnetic constituent and an electrically conductive element enables very efficient heating to occur through both hysteresis loss and Joule effect. The Cu-Co catalyst maintained its remarkable stability for about 50 h throughout the streaming test due to Joule heating. The results offer important insights into the potential of catalysts for RF-assisted chemical processes that do not require a lot of excessive input current or the usage of powerful magnetic metals to attain the optimal chemical configurations. According to Lin Chen et al., who used Co with Ni as an aerogel catalyst, the $CH_4$ conversion in a magnetic-assisted fluidized bed reactor increased by 12 and 7%, respectively, in comparison to a fixed bed reactor and a conventional fluidized bed reactor [80]. This was due to the catalyst's better fluidization reliability. Additionally, over the course of the 50 h reaction, the catalytic performance remained quite steady. The enhanced gas–solid association in the magnetic-assisted fluidized bed reactor and the Co-Ni aerogel catalyst's better catalytic property can both serve as examples of this higher stability.

Attapulgite-derived MFI (ADM) zeolite-encased Ni-Co alloys were developed by Defang Liang et al. utilizing a one-pot technique [81]. The 24 h of dry reforming of methane effectiveness of the 10Ni@ADM, 10Ni@ADM-0.1, and 10NilCo@ADM-0.1 catalysts were investigated at 700 °C at 20/20/60 mL/min of $CH_4/CO_2/N_2$, and the results are displayed in Figure 11a. This investigation was performed to study the influence of the structures of ADM-encased Ni-Co alloys on catalytic stability and activity. The 10Ni1Co@ADM-0.1 catalyst has the highest dry reforming of methane activity, as shown in Figure 11a–c. The stability test results are shown in Figure 11d,e. The primary conversion of $CH_4$ and $CO_2$ by the 10Ni@ADM catalyst was 52 and 95%, respectively, with an $H_2/CO$ molar ratio of 0.88. Furthermore, the 10Ni@ADM catalyst underwent a modest deactivation after 24 h of reaction, as the rates of $CH_4$, $CO_2$ conversions, and $H_2/CO$ molar ratio declined to 45%, 85%, and 0.88, respectively. This might be a result of the catalyst being exposed to high temperatures for a long time, which caused the encased metal grains to migrate to the interface of the zeolite, causing partial metallic Ni sintering and active area loss. It was found that the initial activity of the 10Ni@ADM-0.1 catalyst and the 10Ni@ADM catalyst was related. However, once the reaction was over, the $CH_4$ and $CO_2$ conversions only dropped to 51% and 89%, respectively, showing that the inclusion of NaOH helped increase the catalyst's reliability. In terms of catalysts, the 10Ni1Co@ADM-0.1 catalyst showed the best catalytic performance, achieving the highest initial $CH_4$ and $CO_2$ conversions of 71 and 93%, respectively, and maintaining those conversions throughout the dry reforming of methane reaction. Two mechanisms, including the presence of Co as new active sites that promote $CH_4$ activation/adsorption and the alloy interaction between Ni and Co that promotes $CH_4$ breakdown, may be responsible for the higher initial $CH_4$ conversion when compared to the 10Ni@ADM catalyst. In contrast to the 10Ni1Co/ADM-0.1 catalyst, as shown in Figure 11f, the 10Ni1Co@ADM-0.1 catalyst experiences little degradation over 100 h of the dry reforming of methane reaction, further demonstrating the structural resilience of the ADM zeolite-encased Ni-Co alloys.

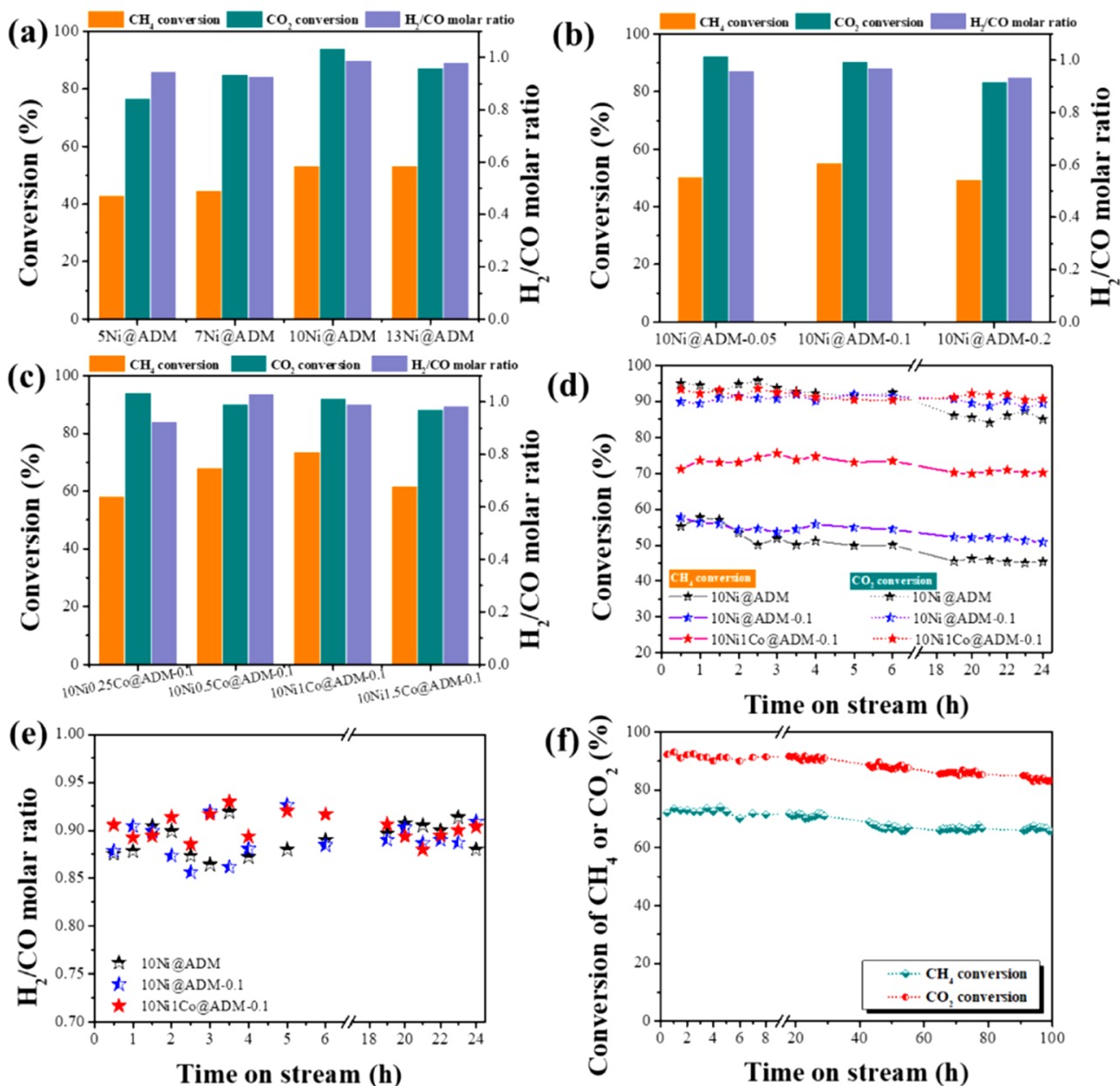

**Figure 11.** Dry reforming of methane reactions for 4 h over the (**a**) xNi@ADM (x = 5–13), (**b**) 10Ni@ADM-y (y = 0.05–0.2), and (**c**) 10NizCo@ADM-0.1 (z = 0.05–1.5) catalysts. (**d**) Conversions of $CH_4$ and $CO_2$ and (**e**) $H_2$/CO molar ratio on the attapulgite-derived MFI (ADM) zeolite-encased Ni-Co alloys. (**f**) Dry reforming of methane reactions for the 100 h test over the 10Ni1Co@ADM-0.1 catalyst. Reproduced from [81]. Copyright (2023), with permission from Elsevier.

Furthermore, the mechanism for dry reforming over the 10Ni1Co@ADM-0.1 catalyst was proposed by Defang Liang et al. [81]. By doping into the metallic Ni lattice of the 10Ni1Co@ADM-0.1 catalyst, the inclusion of Co improved the diffusion of metallic grains and aided in the emergence of Ni-Co alloys. Additionally, the generated Ni-Co alloy interfaces promoted the transfer of electrons from Co to Ni, which aided in the production of electron-rich Ni. This led to the breakdown of C-H bonds in $CH_4$ molecules, which generated the reactive intermediaries CHx*/C* and H* (Figure 12).

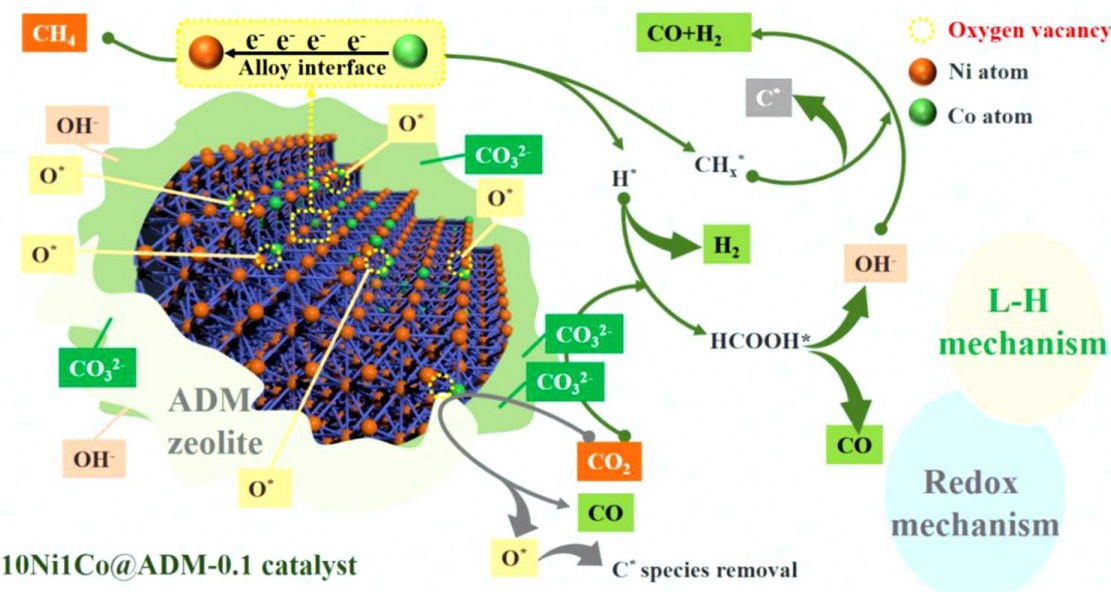

**Figure 12.** Synergetic effect between the attapulgite-derived MFI encapsulation layers and the Ni-Co alloys for the dry reforming of methane reaction on the 10Ni1Co@ADM − 0.1 catalyst. Reproduced from [81]. Copyright (2023), with permission from Elsevier.

Rahman Daiyan et al. show that the Co@CoNC-900 electrocatalyst, which is coated with a single Co atom and enclosed in a graphitic carbon shell, can reliably produce syngas at moderate overpotentials during $CO_2$ reduction processes [82]. They examined the catalyst's linear sweep voltammetry curves and the $CO_2$ reduction process. Figure 13a,b show that as the annealing temperature is raised, and j marginally decreases, which may be connected to the catalyst's depleting Co-$N_4$ centers. Additionally, these findings suggest that the reaction route is not affected by the catalysts' growing surface areas (as the surface area rises when increasing the calcination temperatures). The findings of the bulk $CO_2$ electrolysis support their design strategy since an increase in the annealing temperature, i.e., from 800 °C to 1000 °C, improves $FE_{CO}$ (Figure 13b), while suppressing $FE_{H_2}$ (Figure 13c). The $FE_{CO}$ obtained with the Co@CoNC-800, Co@CoNC-900, and Co@CoNC-1000 catalysts is 36, 45, and 61%, respectively, whereas the $FE_{H_2}$ obtained with these catalysts at the same potential is 59, 43, and 40%, respectively. The combination of a decline in Co-$N_4$ moieties and the potential impact of rising defects on the $CO_2$ reaction selectivity can be used to directly correlate this enhanced selectivity with the $CO_2$ reduction reaction. The syngas ratio provides a vivid demonstration of the adjustment of reaction selectivity (Figure 13d). The authors reported $H_2$:CO ratios of 1.5, 1, and 0.5 for the Co@CoNC-800, Co@CoNC-900, and Co@CoNC-1000 catalysts, respectively, by adjusting the annealing temperature.

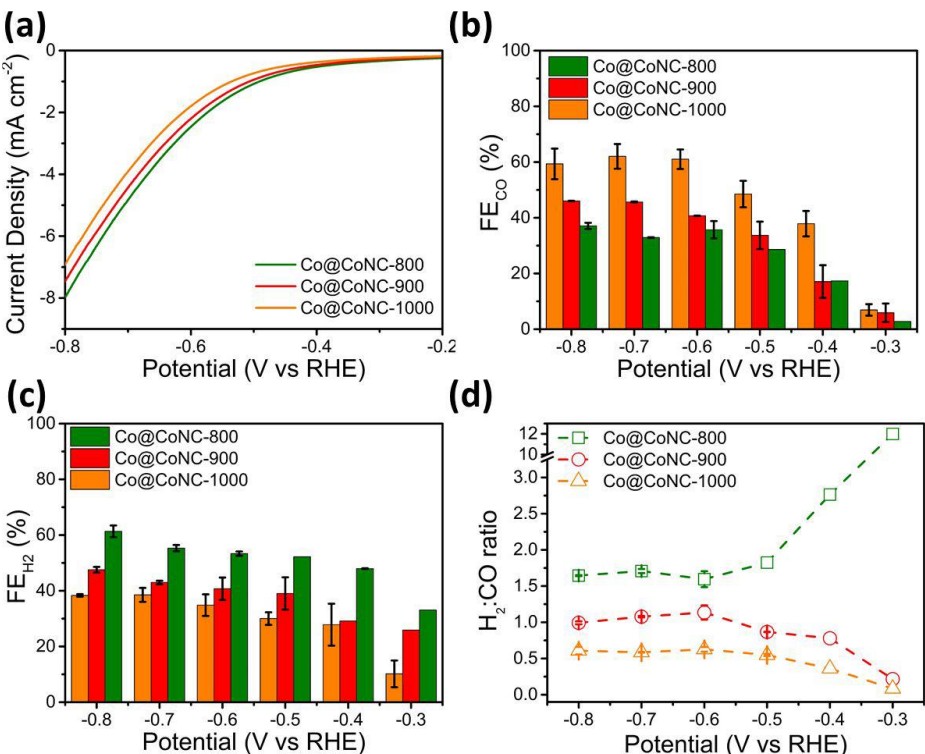

**Figure 13.** Modification of syngas ratio using active site manipulation. (**a**) Linear sweep voltammetry, (**b**) FE_CO, (**c**) FE_H2, and (**d**) H$_2$/CO ratio with applied potential for the prepared catalysts. Reproduced from [82]. Copyright (2020), with permission from the American Chemical Society.

Some of the recent progresses on syngas production/conversion by utilizing magnetic nanomaterials are compared and shown in Table 1.

**Table 1.** Various magnetic materials and their utilization for syngas production/conversion.

| Magnetic Catalysts | Preparation Process | Process | Performance | Reference |
|---|---|---|---|---|
| MgO/MgFe$_2$O$_4$ | Combustion process | Biodiesel production reaction from vegetable oil | Lowest conversion is 82.4%, and maximum conversion is 91.2%. | [83] |
| Ni-Fe | Power-to-gas (hydrogenation method) | CO$_2$ methanation | Conversion of 80% of CO$_2$ to methane at 150 °C. CH$_4$ selectivity of CO$_2$ methanation is 95%. | [84] |
| (Fe, Cu, and K) and rice husk char | Pyrolysis method | Catalytic tar conversion for improving the yield of syngas | The tar conversion efficiency is obtained at 77.1% for RHC, 82.7% for K-RHC, and for 92.6% Fe-RHC at the reforming temperature of 800 °C. | [85] |
| FeSo$_4$ | Steam gasification and pyrolysis process | H$_2$-rich syngas production | The maximum overall H$_2$ yield and exergy efficiency for producing H$_2$ are estimated to be 43.63%. | [86] |
| Ni/A-A-CFA | Biogas dry reforming method | Syngas production | It exhibits the best catalytic activity at CH$_4$ and CO$_2$ conversion rate > 95%. | [87] |
| Fe$_2$O$_3$/Al$_2$O$_3$ | Chemical looping dry reforming process | For CO$_2$ production of hydrogen and syngas | Dry reforming stage for CH$_4$ conversion is 3.84, and syngas yield is 98.32%. | [88] |

**Table 1.** *Cont.*

| Magnetic Catalysts | Preparation Process | Process | Performance | Reference |
|---|---|---|---|---|
| $TiO_2/Fe$ | Fermentation process | For biohydrogen production | Yield increases by 24.9%. | [89] |
| $Fe_3O_4$ and $Fe_5O_{12}$ | Hydrolysis | Biomass production | A 90% harvesting efficiency is obtained. | [90] |
| $Fe_3O_4/CuO$ | Coprecipitation method | Decolorization of water | A high dye removal efficiency of 94 % is achieved with 20 wt% $Fe_3O_4/CuO$ composition at ambient conditions and a reaction time of 90 min. | [91] |
| Ni, Ni-Co, Ni-Fe, and Ni-Cu | Hydrothermal synthesis | Syngas production and carbon bio nanofilament | The highest sustainability factor (0.66) and carbon yield (424%) are obtained. | [17] |
| Ni/Al | Coprecipitation method | Production of hydrogen-rich syngas from biomass pyrolysis | Ni/Al-700 catalyst can increase gas yield by 30–80%. | [92] |
| $Fe_2O_3$ and $MgFe_2O$ | Pyrolysis | Syngas production from biomass | Tar conversion efficiency reaches 94.1% with a high gas yield of 493.5 mL/g. | [93] |
| $Fe/Ca_xO$ | Simple precipitation method | Pyrolysis - gasification used for syngas production and tar removal | At an optimized composition of Ca/Fe 2/1, gasification yield efficiency (76.4%) is obtained. | [94] |
| Porous Ni, Ni-Co, Ni-Fe, and Ni-Cu | Precipitation method | Biomass decomposition to produce syngas | A high carbon yield efficiency of 36.43% is obtained. | [17] |
| Ni/Al | Ultrasonic-assisted incipient wetness impregnation method | Catalytic conversion of tar to syngas | With the addition of a catalyst, $H_2$ increases to 146%. | [95] |
| $Ni-Cu/Al_2O_3$ | Impregnation method | Syngas production by methanol steam reforming | An increase in Ni content results in an increase in CO and a decrease in $CO_2$ yields. | [96] |
| Sc@ Ni/Fe | Pyrolysis method | Impregnation of biomass | At 600 °C, the conversion efficiency reaches 90.07%. | [97] |
| Fe-Ni/CNF | Pyrolysis method | Syngas production from pyrolysis gasification of biomass and plastic waste | Conversion efficiency of 87.90%. | [98] |
| Ni/Ru | Steam reforming process | Ru catalyst favors $H_2$ and CO production | 90% | [99] |

## 7. Effect of External Magnetic Field

We have discussed Fe-, Ni-, and Co-based magnetic materials for their utilization in syngas production/conversion in the previous section. This section discusses the effects of external magnetic field on magnetic nanomaterials during synthesis and applications. The effects of an external magnetic field could alter the charge transfer process during applications [100,101]. By transmitting loads of energy to the atomic and molecular dimensions of the materials, an external magnetic field can influence the microstructure and material characteristics. The production of static magnetic energy by a magnetic field affects the free energy of substances during synthesis and has an impact on their nucleation selectivity and rate, which is linked to metastable elements and innovative materials. Modern electrocatalysts can now be generated using magnetic fields, which are associated with nucleation and growth, as well as phase modulation. By enhancing the mass transfer on the working electrode, the magnetic effects of an external magnetic field, such as the magnetothermal effect, magnetohydrodynamic and micro-magnetohydrodynamic effects, Maxwell stress, and Kelvin force with spin selection effect, could indeed alter the reaction trends. This is advantageous for increasing the electrocatalytic performance of electrocatalysts with the potential utilization of external magnetic fields [102]. Various synthesis processes and

applications have proven that an external magnetic field could improve the efficiency of materials, depending on their magnetic characteristics [103,104].

An external magnetic field-coupling microfluidic synthesis process consisting of a reaction liquid preheater, a reaction-nucleation stage heating and temperature controller, a thermostat, a low-temperature abrupt termination collector, and a MF controller could be utilized for the synthesis process [105]. According to Junmei Wang et al., the synthesis of a $Fe_2$ Pt/C catalyst under an external magnetic field has higher catalytic performance. A concurrently applied 1.4 T external magnetic field coupled with the microfluidic process results in $Fe_2$ Pt/C nanocrystals of larger size. This method gives better catalytic performance on both ethanol and methanol oxidation reactions compared to the samples synthesized without a magnetic field [106]. Similarly, He et al. demonstrated that distinct ferric sulfide minerals, such as greigite and marcasite, have different magnetic properties, including coercive force, remnant, saturation, and magnetization. By using an in situ MF-assisted hydrothermal process to synthesize the ferric sulfide minerals, they were able to achieve this [107].

Figure 14 shows the four steps in the magnetic field-coupling microfluidic preparation process, including the preparation of the precursor, the preheating of the precursor, the nucleation of QDs, and the collection of QDs. The magnetic field along the channel was created using the solenoid electromagnet. The products were collected under $N_2$ gas. Xiaoxiong Zhao et al. concluded that the magnetic fields had a significant impact on the nanoparticles' magnetic and optical properties. Regulation of the magnetic field was conveniently realized during the synthesis process. Co-doped ZnSe QD aggregates with a good size distribution were produced using a simplified microfluidic technique and a customizable magnetic field. These aggregates were created by numerous tiny nanoparticles (with diameters of about 4–6 nm) fusing together. Eventually, the authors insisted that by controlling the applied magnetic fields, particle sizes, and Co doping amounts, the Mr, Ms, Hc, net magnetization, and ferromagnetism of the QDs were adjusted. To alter the distinct magnetic characteristics of the QDs, one might alter the ratio of ferromagnetic to antiferromagnetic phases. It was realized that adding Co and altering the doping amount modified the band gaps of the QDs when estimating the band gap through using the UV-vis absorbance and R%.

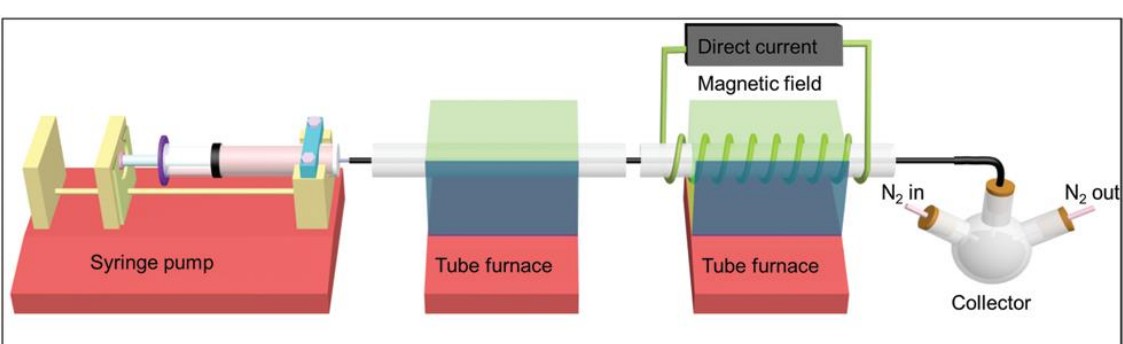

**Figure 14.** The four steps in the magnetic field-coupling microfluidic preparation process. Used with permission from [105]; permission conveyed through Copyright Clearance Center, Inc.

Po-Wei Lan et al. investigated various Ni/Fe catalyst ratios for increased carbon dioxide methanation by employing chemical reduction with an external magnetic field, and they showed that the $Ni_8Fe_2$ catalyst could convert more than 80% of $CO_2$ to methane at 150 °C. Furthermore, 95% of $CO_2$ methanation's selectivity was for $CH_4$ [84]. Figure 15 shows the experimental process of the magnetic field while synthesizing nanomaterials. Chemical reduction was cast off to develop a sequence of Ni/Fe catalysts with varying Fe/Ni ratios ($Ni_9Fe_1$, $Ni_8Fe_2$, and $Ni_7Fe_3$). The resulting solution was then transferred to a magnetic field reactor and heated to 80 °C while being subjected to a 500 G magnetic field. A neodymium magnet performed well enough to harvest the generated particles,

which were subsequently purified by being washed with deionized water numerous times. Before being used, the produced Ni/Fe catalysts with various Ni/Fe ratios were dried in a desiccator.

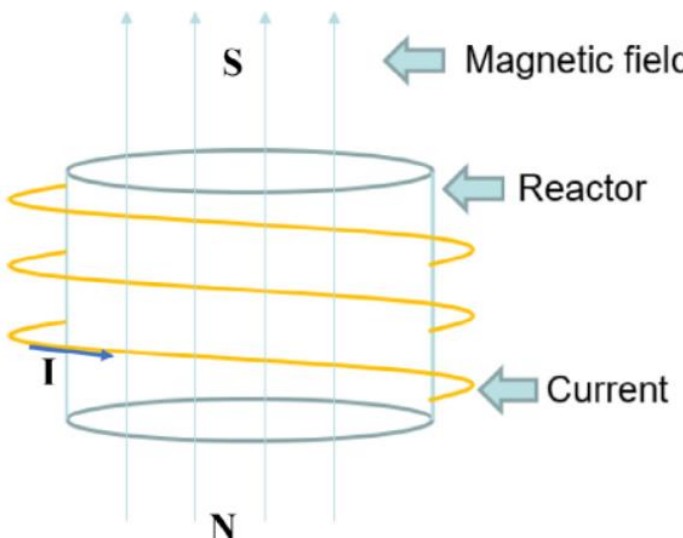

**Figure 15.** The magnetic field reactor for Ni-Fe catalyst preparation. Reprinted from [84], Copyright (2021), with permission from Elsevier.

Due to the simultaneous endothermic $CO_2$ reformation and exothermic partial oxidation of methane to syngas in the occurrence of oxygen, the methane-to-syngas transition process takes place in an energy-efficient manner, demanding little or no external source of energy. When Fe is injected into zeolite substances, Fe-containing acidic sites, iron oxides, and numerous paramagnetic O ions are all generated. The improved efficiency in a magnetic field is closely related to the magnetically triggered Fe ions and surface paramagnetic oxygen [108]. Methyl-functionalized silica and methyl-functionalized cobalt ferrite–silica nanoparticles were evaluated for their capacity to increase the generation of bioethanol in syngas fermentation via Clostridium ljungdahlii [109]. The $CoFe_2O_4/SiO_2/CH_3$ nanoparticles demonstrated higher enhancement of syngas mass transfer when these nanoparticles were utilized to promote the syngas–water mass transfer. It was verified that the nanoparticles' ability to increase mass transfer was preserved after being recovered using a magnet and employed five more times. When utilizing both varieties of nanoparticles, the syngas fermentation process produced more biomass, ethanol, and acetic acid. Due to the increased syngas mass transfer, the $CoFe_2O_4/SiO_2/CH_3$ nanoparticles were more successful for syngas fermentation. The inclusion of $CoFe_2O_4/SiO_2/CH_3$ nanoparticles boosted the generation of biomass, ethanol, and acetic acid by around 228%, 214%, and 60%, respectively, compared to a control. By using recovered nanoparticles for fermentation, the nanoparticles' reusability was demonstrated, and the results clearly show the advantages of utilizing magnetic nanomaterials in syngas production.

Using an external MF-coupling microfluidic method, homogeneous ultrasmall-sized $Fe_2Pt/C$ nanocatalysts are synthesized [106]. The reactants could be controlled by the magnetizing force linked to the magnetic field gradient and the Lorentz force when an external magnetic field is applied. In turn, their catalytic performances in the electrochemical oxidation of methanol and ethanol can be considerably enhanced by controlling the size, composition, electronic status, and magnetic characteristics of the catalysts. This investigation can help us better understand how an external magnetic field influences nanocatalyst synthesis and offer us a good concept for enhancing the catalytic efficiency of electrochemical nanocatalysts. An attempt was made to determine whether it is possible to increase the catalytic performance of a Fe-based catalyst for the removal of $AsH_3$ by using a low-energy external magnetic field [110]. With a 0.1 T magnetic field present and

a temperature of 120 °C, the catalytic performance for $AsH_3$ degradation was improved by 52%. Fe-containing acidic sites, iron oxides, and many paramagnetic O ions were all introduced when Fe was incorporated into zeolite materials. The magnetically activated Fe species and surface paramagnetic oxygen were substantially linked to the increased efficiency in a magnetic field. When a considerable magnetic field (450 mT) was introduced to the anode, electrocatalytic water oxidation in the alkaline media was revealed to be significantly strengthened by magnetism [108]. The use of an external magnetic field for water splitting investigations is suggested by the fact that magnetic enhancement works even on decorated Ni-foam electrodes with extremely high current densities, increasing their intrinsic activity by around 40% to reach over 1 $A/cm^2$ at low overpotentials. Thermal conversion investigations were carried out using magnetic field-assisted catalytic pyrolysis of biomass to produce $H_2$-rich gases from wood sawdust [111]. The outcomes of this experiment revealed that a 10 weight percent of a Ni/CaO catalyst with adequate magnetic properties had a favorable catalytic capacity to create $H_2$ by pyrolyzing biomass. With increased magnetic field strength, the $H_2$ quantity and output increased. The $H_2$ content and output exceeded 62 vol.% and 469 mL/g, respectively, at 650 °C, 80 m T magnetic field intensity, and 10 wt.% Ni/CaO catalyst. They improved by around 10 % and 20 %, correspondingly, in comparison to the absence of a magnetic field. As a result, a promising approach for using biomass is biomass catalytic pyrolysis supported by a magnetic field.

An external magnetic field was connected with a catalytic packed bed reactor for selective generation of methanol via $CO_2$ hydrogenation over Cu and Fe loaded on highly porous ZSM-5 zeolite in an effort to address both economical and environmental concerns regarding the green generation of methanol [88,112]. It was observed that the selectivity and catalytic activity of $CO_2$ hydrogenation significantly improved with the presence of an external magnetic field. The catalyst 10Cu-10Fe/ZSM-5 showed the maximum $CO_2$ conversion at a $CO_2/H_2$ molar ratio of 1:3 in the presence of an external magnetic field. At 220 °C, the external magnetic field enhanced the selectivity to methanol and the $CO_2$ conversion by factors of 1.7 and 2.24, respectively. Therefore, the use of magnetic fields promotes $CO_2$ adsorption and results in the generation of selective methanol, paving the way for the potential to introduce a green and sustainable technology in both petrochemical and chemical processes. An external magnetic field is used in an initiative to improve the catalytic efficiency of the $CO_2$ hydrogenation reaction based on green and efficient exploitation concepts [6]. Based on the activity and selectivity of Fe-based catalysts with ferro/ferrimagnetic properties, the effects of magnetic field orientation and magnetic flux density were demonstrated. In comparison to those without a magnetic field, $CO_2$ conversions were dramatically increased by 1.5–1.8 times with an external magnetic field, particularly in the north–south (N–S) direction, while activation energy was reduced by 1.1–1.15 times. When the magnetic flux density was changed, the rate of $CO_2$ conversion improved in the following order: 27.7 mT > 25.1 mT > 20.8 mT. These exceptional catalytic activities can be attributed to the magnetic field's role in facilitating reactant adsorption and surface reactions over the magnetized Fe catalysts, which reduces apparent activation energy and increases selectivity to hydrocarbons and $CH_3OH$.

Successful demonstrations of a magnetic field's impact on $CO_2$ conversion over the Cu-ZnO/$ZrO_2$ catalyst in the hydrogenation reaction were established [113]. The maximum $CO_2$ conversions were achieved at 220 °C with a magnetic field of 20.8 mT in the SN direction, which was three times greater than the results obtained without a magnetic field. Haiping Pan et al. established efficient magnetic field adjustment of the radical-pair spin states in electrocatalytic carbon dioxide reduction [114]. This work demonstrates that applying an external magnetic field for electrocatalytic $CO_2$ reduction to formate/formic acid considerably increases the catalytic performance of tin nanoparticle catalysts. The generation of formic acid can roughly double in comparison to zero magnetic fields when a standard Sn nanoparticle electrode is used as an example, and a constant external magnetic field of about 0.9 T is utilized. Figure 16 shows the effect of the external magnetic field on the formate yield, which depends on the applied electrode potential for different molar

concentrations of $KHCO_3$ electrolytes at 1.7 V (vs. Ag/AgCl). This discovery paves the path for increased formate synthesis in the electrocatalytic reduction of $CO_2$ and points to the benefits of radical-pair spin states in the electron transfer process.

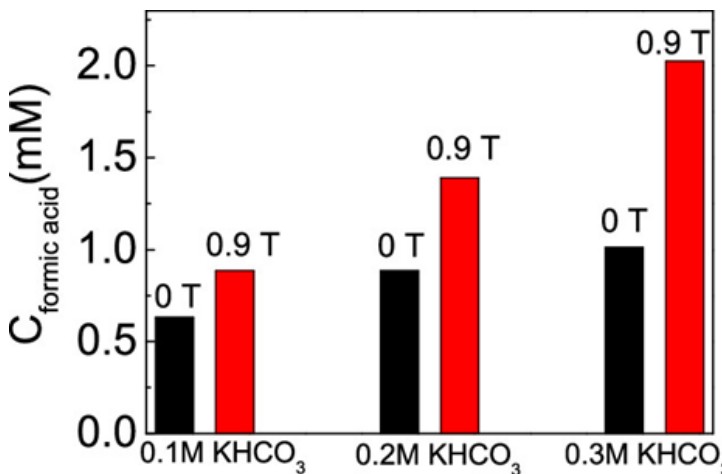

**Figure 16.** Magnetic field-dependent enhancement of formate yield ($-1.7$ V (vs/ Ag/AgCl)) with the same applied electrode potential for different concentrations of $KHCO_3$ electrolytes. Reprinted with permission from [114]. Copyright (2020), with permission from the American Chemical Society.

Magnetic field-assisted acceleration of $CO_2$ reforming of methane over an innovative hierarchical Co/MgO catalyst was performed [115]. The Co/MgO catalyst was developed through a hydrothermal precipitation route. The $CO_2$-$CH_4$ reforming reaction was carried out at atmospheric pressure in a bench-scale fluidized bed reactor system that included a gas feeding functionality, a fluidized bed reactor colocated with a heat furnace, and a GC analyzer. The catalyst nanoparticles were pre-reduced by utilizing $H_2$ (100 mL/min) at 800 °C for 1 h before the experiment. Following the reduction, the feed gas was fed into the reactor to begin the $CO_2$ reforming test, with a $CH_4/CO_2/N_2$ molar ratio of 1:1:1. A total of 0.2 g of the nanoparticles was added to the fluidized bed reactor for the activity. Four parallel solenoids created an axial uniform magnetic field in the case of the MFB reactor, and the current of the power source could be adjusted to modulate the magnetic field's strength (H). The magnetic field employed had a 130 Oe intensity, which is equivalent to a 344W electric energy input. The calcined CoO/MgO particle has a hierarchical architecture and gyroscopic-like structure, with an average dimension of 25 m. Many octahedral MgO nanostructures, with a crystal size of around 40 nm, self-assembled to form the particle. Figure 17 illustrates the stability test results for the Co-MgO catalyst with the MFB and CFB reactors. The first $CO_2$-$CH_4$ reformation performance of the Co/MgO catalyst in the CFB reactor was outstanding, with high $CO_2$ conversion (about 92%) and $CH_4$ conversion (nearly 90%). The conversions of $CO_2$ and $CH_4$ under the MFB and CFB reactors did not significantly decrease during the stability test of 24 h (Figure 17a,b). For the first three hours of the course of the reaction, the production of $H_2$ and CO for the MFB reactor displayed a modest induction period before remaining constant (Figure 17c,d). The conversions of $CO_2$ and $CH_4$ were enhanced by 22% and 30%, respectively, in the magnetic-assisted fluidized bed reactor at 800 °C, with high gas at an hourly space velocity of $15 \times 10^4$/h. The hierarchical Co/MgO catalyst's increased activity and great resistance to coke deposition are related to the development of clusters, which exhibit a high frequency of breaking and consolidating under the influence of a magnetic field.

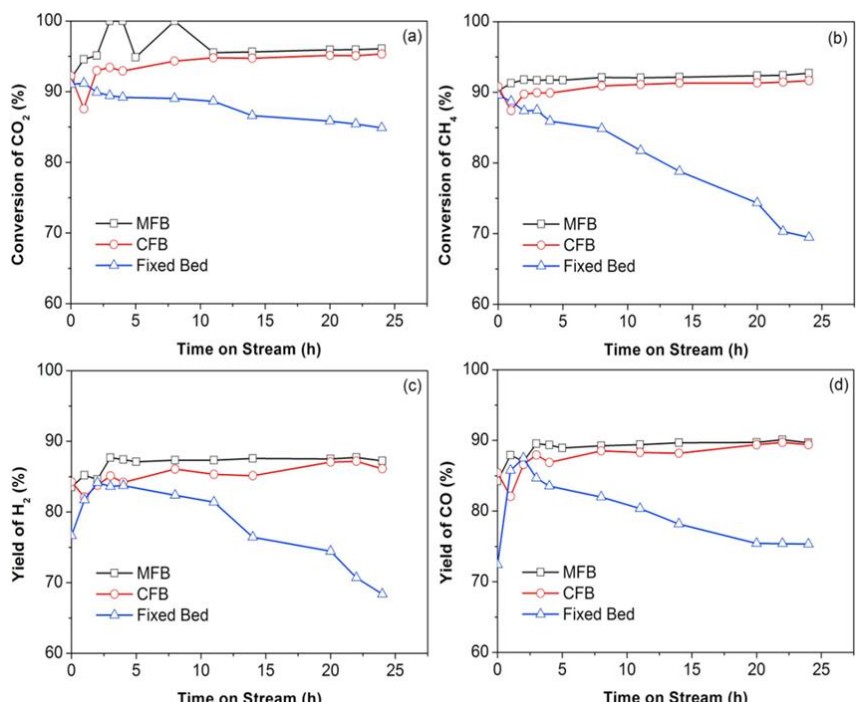

**Figure 17.** Catalytic stability experiments with GHSV of 60,000/ h. Conversions of (**a**) $CO_2$, (**b**) $CH_4$, the yields of (**c**) $H_2$ and (**d**) CO. Reprinted from [115]. Copyright (2018), with permission from Elsevier.

In general, carbon development can originate from the Boudouard reaction below 700 °C, where CO dissociates to generate surface carbon and $CO_2$, and $CH_4$ decomposition occurs beyond 557 °C, where $CH_4$ totally dissociates to form solid carbon on the surface of the catalyst and produce $H_2$. After 24 h of $CO_2$-$CH_4$ reforming at 800 °C, the used catalysts were examined by SEM and TEM to investigate the carbon deposition and changes to the catalysts' crystal size. Based on the structural examination, carbon diffraction peaks are clearly visible from the used catalysts in the CFB and MFB reactors, suggesting the presence of carbon deposition. As can be seen from the SEM image (Figure 18a–c), neither the CFB nor the MFB reactors contain any long carbon nanotubes or wires on top of the Co/MgO particles. In the conventional fluidized bed, numerous short, irregular carbon nanotubes are seen deposited on the exterior of the used catalyst in comparison to the fresh catalyst. On the catalyst that had been used up in the MFB reactor, less carbon nanotubes are seen. The fresh Co/MgO catalyst has Co NPs that are less than 5 nm in size, according to the TEM micrograph in Figure 18d. After the stability test, the average Co NP dimensions of the used Co/MgO catalysts for both the CFB and MFB reactors reveal a minor sintering compared to the fresh Co/MgO catalyst. The Co NPs after sintering show no overt response to the presence of magnetic field. This outcome contrasts with that of CO methanation in a magnetic fluidized bed reactor, where the axial magnetic field helped reduce catalyst agglomerates and prevent the sintering of NiCo particles [116]. The hierarchical Co/MgO catalyst's structure and modified particle size, which enable the hierarchical Co/MgO particle to remain an individual particle, are responsible for this variance. Following the stability test in the CFB reactor, numerous Co NPs are detached from the MgO carrier's surface and completely disseminated over the carbon nanotubes. However, due to the magnetic susceptibility of Co NPs and their interaction with one another, this transition of Co NPs is considerably diminished in the MFB reactor, as observed from the TEM micrograph in Figure 18d–f. Additional research is required to better understand the Co NPs' transition phenomena. To conclude, an axial magnetic field promotes the stability of Co particles, reduces carbon deposition, and increases the reforming rate. The suggested method successfully produces syngas via $CO_2$-

CH$_4$ reforming with improved catalytic activity through the combination of a magnetic fluidized bed and a structured nanoparticle catalyst.

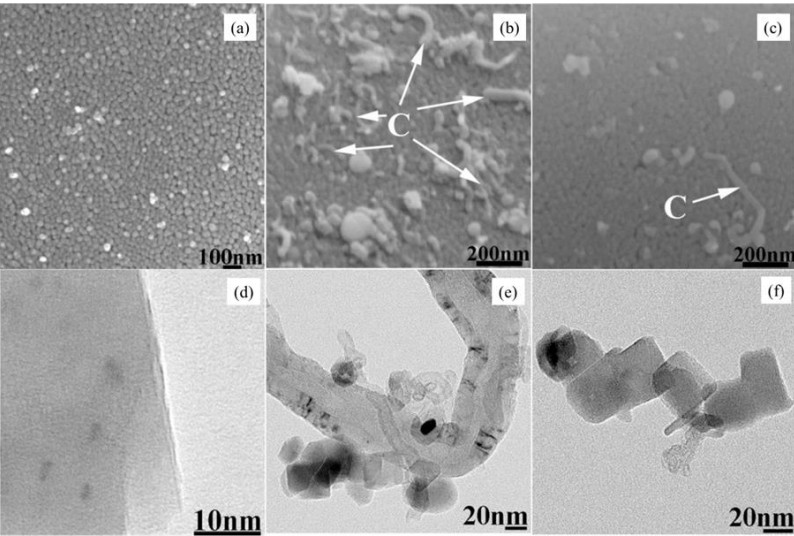

**Figure 18.** (**a**–**c**) SEM and (**d**–**f**) TEM micrographs of pure Co-MgO and used Co-MgO after reforming reaction in a conventional fluidized bed reactor and a magnetic-assisted fluidized bed (MFB) reactor. Reprinted from [115]. Copyright (2018), with permission from Elsevier.

The reviewed studies suggest that there are still a lot of opportunities and challenges in the effective utilization of external magnetic fields for syngas production and conversion applications. It is important to optimize magnetic nanoparticle or nanocomposite formation as the magnetic responses to external magnetic fields mainly depend on the magnetic characteristics of the materials. Furthermore, the range of the external magnetic field for selected magnetic materials also needs an optimization to obtain a better efficiency in syngas production and conversion applications. Moreover, understanding the role of external magnetic field in syngas production and conversion is also crucial, as the magnetic field itself could alter the chemical reaction kinetics even though it does not have magnetic characteristics. We could see that a non-magnetic material could also be influenced by an external magnetic field. By considering these factors, researchers could conduct a better investigation on the effect of external magnetic field on syngas production and conversion to achieve a better efficiency. The main characteristics of magnetic nanomaterials as catalysts, the types of magnetic nanomaterials, and the experimental parameters that can be tuned to improve the performance of catalysts for syngas production/conversion is schematically shown in Figure 19.

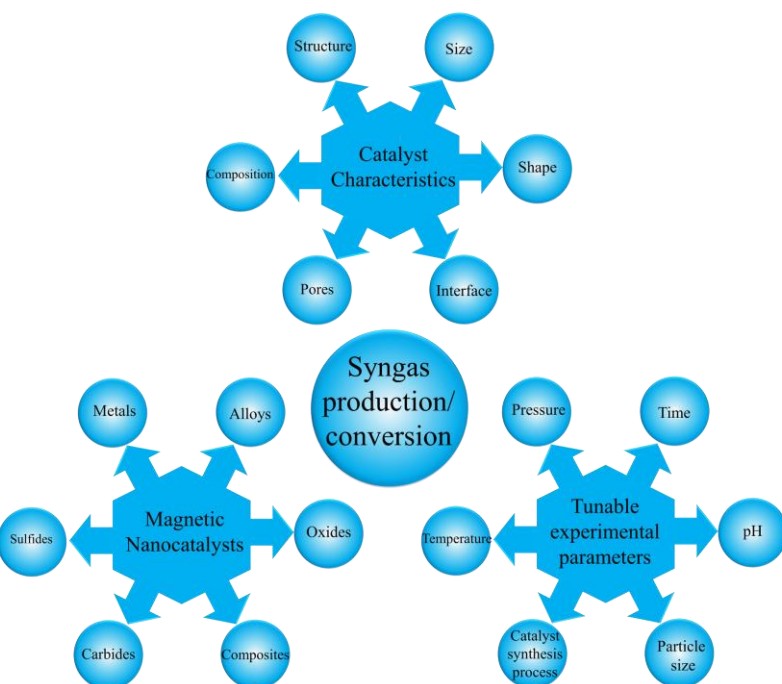

**Figure 19.** Schematic diagram for the typical characteristics of magnetic nanomaterials, types of magnetic nanomaterials, and experimental parameters that can be tuned to improve the performance of catalysts for syngas production/conversion.

## 8. Summary and Scope

We have introduced a brief discussion of magnetic materials, and the different preparation processes for the synthesis of magnetic nanomaterials are also discussed. A brief introduction to the syngas production/conversion process and the major experimental parameters, such as temperature, pH, time, and other reaction parameters that can influence the efficiency of the process, are included. The magnetic materials that this review has mainly focused on belong to Fe-, Ni-, and Co-based materials. The last section discusses the possibility of effectively utilizing an external magnetic field for the synthesis of magnetic nanocatalysts. The role of external magnetic field on the catalysis process of syngas production/conversion process is discussed with suitable examples. The advantages of magnetic nanomaterials over other available catalysts are their magnetic characteristics. One can tune the magnetic properties of magnetic materials by tuning their stoichiometry and morphology. Furthermore, the magnetic properties could be tuned by the addition of foreign elements or materials through composite formation. The required magnetic characteristics of these magnetic materials could be achieved by employing suitable experimental parameters and a suitable surface modifier. The magnetic separation process could be used to collect used magnetic nanocatalysts, which may be used for another cyclic performance. We strongly believe that the addition of an external magnetic field to the synthesis of magnetic nanoparticles and to the catalysis process could effectively alter the efficiency of the process. However, the range of the external magnetic field should be optimized, depending on the magnetic characteristics of the magnetic nanocatalysts for better performance.

**Author Contributions:** Conceptualization, A.T. and N.C.; writing—original draft preparation, S.J.J.K. and S.P.; writing—review and editing, P.S., S.D., S.S., S.-K.K. and R.M.; supervision and writing—review and editing, A.T. and N.C. All authors have read and agreed to the published version of the manuscript.

**Funding:** This study was funded by the Agencia Nacional de Investigacion y Desarollo-SA 77210070.

**Data Availability Statement:** The data presented in this study are available from the corresponding author upon request.

**Conflicts of Interest:** The authors declare no conflict of interest.

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
