# Peer review of "Magnetic Nanomaterials as Catalysts for Syngas Production and Conversion"

_catalysts, doi:10.3390/catal13020440_

Round 1
Reviewer 1 Report
I appreciate the efforts of the author; this review article is suitable for “Catalysis”. However, there are some comments to help the author to reach the standard of “Catalysis”.
1. There are only 12 Figures in this manuscript. A good review article related to magnetic materials for syngas production/conversion normally has 15-20 Figures. Furthermore, the pictures should be in a group of pairs, like 2,4,6,8,10. Please try to adjust 2-5 references in a Figure. Then the review will be outstanding.
2. There is no connection in the discussion. I strongly recommend the author please connect the paragraphs of discussion with each other in a logical way, like “to shed more light on the emerging magnetic materials for syngas production/conversion,” etc.
3. Make sure all abbreviations are written out in full the first time used. This is particularly important in the abstract and the conclusions but work through the entire ms carefully from this perspective.
4. There are grammatical, syntax, or word usage errors in the manuscript. Please improve the English of this manuscript.
5. Review published in “Catalysis” must explain the significant advances provided in the latest approaches and understanding compared to previous literature, and/or demonstrate convincingly potential in new advancements. The Conclusions of your paper are especially important for this. Therefore, please try to sharpen this further. The optimal Conclusion should include:
• A summary of your key points in the review.
• A highlight of the latest approaches, new concepts, and innovations.
• A summary of key improvements compared to findings in the literature [provide a couple of references to indicate key improvements].
• Your vision for future work.
6. After the conclusion, the author should also add 5-7 points related to future prospective.
7. This is a review article and there is no any table comparison for emerging magnetic materials for syngas production/conversion. Please present a comparison of emerging magnetic materials for syngas production/conversion in a scientific table (at least one table with 20-30 refences.
8. Please try to read and cite already published review article in famous journals 10.1016/j.cej.2022.140601, 10.1039/D2DT03272D like these articles published in CEJ.
Author Response
The authors gratefully acknowledge the valuable comments and suggestions given by the reviewers. The authors feel that these comments are very much useful in improving the standard of this manuscript. All the suggestions and corrections given by the reviewers are addressed properly.
Reviewers' comments:
Comments and Suggestions for Authors
I appreciate the efforts of the author; this review article is suitable for “Catalysis”. However, there are some comments to help the author to reach the standard of “Catalysis”.
Comment 1. There are only 12 Figures in this manuscript. A good review article related to magnetic materials for syngas production/conversion normally has 15-20 Figures. Furthermore, the pictures should be in a group of pairs, like 2,4,6,8,10. Please try to adjust 2-5 references in a Figure. Then the review will be outstanding.
Response: The authors agree to the suggestion given by the reviewer. The number of figures has been increased in the revised manuscript as per the reviewer suggestion.
Comment 2. There is no connection in the discussion. I strongly recommend the author please connect the paragraphs of discussion with each other in a logical way, like “to shed more light on the emerging magnetic materials for syngas production/conversion,” etc.
Response: The paragraph of discussion is connected in a logical way as per the reviewer suggestion.
Comment 3. Make sure all abbreviations are written out in full the first time used. This is particularly important in the abstract and the conclusions but work through the entire ms carefully from this perspective.
Response: The abbreviations were checked through out the manuscript and expansion is given when it is used for the first time.
Comment 4. There are grammatical, syntax, or word usage errors in the manuscript. Please improve the English of this manuscript.
Response: The spelling and grammar were checked in the revised article.
Comment 5. Review published in “Catalysis” must explain the significant advances provided in the latest approaches and understanding compared to previous literature, and/or demonstrate convincingly potential in new advancements. The Conclusions of your paper are especially important for this. Therefore, please try to sharpen this further. The optimal Conclusion should include: A summary of your key points in the review, A highlight of the latest approaches, new concepts, and innovations., A summary of key improvements compared to findings in the literature [provide a couple of references to indicate key improvements] and Your vision for future work.
Response:
The recent advance from the available literature is included. The conclusions were modified as per the reviewer suggestion. The future direction and scope of this work is suggested in the conclusion.
Comment 6. After the conclusion, the author should also add 5-7 points related to future prospective.
Response:
The future perspective is added in next to the conclusion section.
Comment 7. This is a review article and there is no any table comparison for emerging magnetic materials for syngas production/conversion. Please present a comparison of emerging magnetic materials for syngas production/conversion in a scientific table (at least one table with 20-30 refences.
Response: The authors agree to the remark mentioned by the reviewer. The table related to the magnetic materials as catalysts for the syngas production and conversion and its efficiency has been tabulated in the revised article.
Comment 8. Please try to read and cite already published review article in famous journals 10.1016/j.cej.2022.140601, 10.1039/D2DT03272D like these articles published in CEJ.
Response:
The suggested articles were included in the revised manuscript.
Reviewer 2 Report
The review article on “Magnetic Nanomaterials as Catalysts for Syngas Production and Conversion” is well-written and interesting. The authors specifically focused on the Fe, Ni, and Co-based magnetic materials for syngas production and conversions. The last section on the effect of magnetic field on the synthesis and applications of the magnetic field will be interesting for the readers. The review article may be considered for publication after minor revision. I would like to suggest a few corrections before it gets acceptance for the improvement of the article.
1. The reference for sections 1-3 may be improved. Recent references on syngas and magnetic nanomaterials may be included.
2. In Some places, it is highlighted with color, for example, in figure 6. The authors should send the corrected version of this manuscript during revision.
3. In lines 244-255, the reference should be added for wand et al.
4. In line 386, a reference should be added for Zhang et al.
5. In line 585, a reference should be added for Jiao Zhao et al.
6. In line 631, a reference should be added for M.V. Tsodikov et al.
7. The % is written as percent in some places and written as % in a few places. The author should follow the same pattern throughout the manuscript.
8. Similarly, Fe and Ni are written as iron and Nickel in some places. The author should introduce the acronym at the very first. After that, they may use the acronym alone for the next sections, instead of using expansion.
Author Response
The authors gratefully acknowledge the valuable comments and suggestions given by the reviewers. The authors feel that these comments are very much useful in improving the standard of this manuscript. All the suggestions and corrections given by the reviewers are addressed properly.
Reviewers' comments:
Comments and Suggestions for Authors
The review article on “Magnetic Nanomaterials as Catalysts for Syngas Production and Conversion” is well-written and interesting. The authors specifically focused on the Fe, Ni, and Co-based magnetic materials for syngas production and conversions. The last section on the effect of magnetic field on the synthesis and applications of the magnetic field will be interesting for the readers. The review article may be considered for publication after minor revision. I would like to suggest a few corrections before it gets acceptance for the improvement of the article.
Comment 1. The reference for sections 1-3 may be improved. Recent references on syngas and magnetic nanomaterials may be included.
Response:
The reference sections have been improved by the addition of recent articles in the revised manuscript.
Comment 2. In Some places, it is highlighted with color, for example, in figure 6. The authors should send the corrected version of this manuscript during revision.
Response:
The errors have been corrected in the revised manuscript.
Comment 3. In lines 244-255, the reference should be added for wand et al.
Response:
The reference has been added in the revised manuscript.
Comment 4. In line 386, a reference should be added for Zhang et al.
Response:
The reference has been added in the revised manuscript.
Comment 5. In line 585, a reference should be added for Jiao Zhao et al.
Response:
The reference has been added in the revised manuscript.
Comment 6. In line 631, a reference should be added for M.V. Tsodikov et al.
Response:
The reference has been added in the revised manuscript.
Comment 7. The % is written as percent in some places and written as % in a few places. The author should follow the same pattern throughout the manuscript.
Response:
The suggested corrections were included in the revised manuscript.
Comment 8. Similarly, Fe and Ni are written as iron and Nickel in some places. The author should introduce the acronym at the very first. After that, they may use the acronym alone for the next sections, instead of using expansion.
Response:
The suggested corrections were included in the revised manuscript. The acronyms were checked.
Reviewer 3 Report
The production and conversion of syngas has always been an important research focus and hotspot in the field of energy and chemical industry. The design, preparation and research and development of solid catalysts are the most critical. In this manuscript, the authors summarize the synthesis processes of various magnetic nanomaterials and their composites that could be utilized as catalysts for syngas production and conversion. The possible influence of magnetic characteristics of the magnetic nanomaterials with an external magnetic field also discussed. I think this review is referential to researcher in this field. I suggest this manuscript be accepted after a minor revision.
(1) In Figure 6, the authors should write the subscript of the chemical formula according to the standard.
(2) In the part of tuning of experimental parameters, on page 14, the particle size is not part of the tuning experimental parameters. The particle size is only a result of control. It cannot be juxtaposed with temperature, pressure and pH value.
(3) During the catalytic reaction process, the morphology of the catalyst plays an important role for the catalytic activity. Currently, some core-shelled and hollow porous structures are extensively used in catalysis, and the following papers are encouraged to be cited: Adv. Funct. Mater., 2019, 29, 1806588 (doi: 10.1002/adfm.201806588). ChemistrySelect, 2022, 7, e202202258 (doi.org/10.1002/slct.202202258)
(4) In the conclusion part, some prospects need to be added about magnetic nanometer catalyst for syngas production and conversion.
Author Response
The authors gratefully acknowledge the valuable comments and suggestions given by the reviewers. The authors feel that these comments are very much useful in improving the standard of this manuscript. All the suggestions and corrections given by the reviewers are addressed properly.
Reviewers' comments:
Comments and Suggestions for Authors
The production and conversion of syngas has always been an important research focus and hotspot in the field of energy and chemical industry. The design, preparation and research and development of solid catalysts are the most critical. In this manuscript, the authors summarize the synthesis processes of various magnetic nanomaterials and their composites that could be utilized as catalysts for syngas production and conversion. The possible influence of magnetic characteristics of the magnetic nanomaterials with an external magnetic field also discussed. I think this review is referential to researcher in this field. I suggest this manuscript be accepted after a minor revision.
Comment (1) In Figure 6, the authors should write the subscript of the chemical formula according to the standard.
Response:
We have used the figures from the published articles. As the figures were adopted from the article, we did not change the figures. Anyway, now we have modified the figures as per the reviewer suggestion.
Comment (2) In the part of tuning of experimental parameters, on page 14, the particle size is not part of the tuning experimental parameters. The particle size is only a result of control. It cannot be juxtaposed with temperature, pressure and pH value.
Response:
We respect the reviewer suggestion, We have removed the section on the particle size.
Comment (3) During the catalytic reaction process, the morphology of the catalyst plays an important role for the catalytic activity. Currently, some core-shelled and hollow porous structures are extensively used in catalysis, and the following papers are encouraged to be cited: Adv. Funct. Mater., 2019, 29, 1806588 (doi: 10.1002/adfm.201806588). Chemistry Select, 2022, 7, e202202258 (doi.org/10.1002/slct.202202258)
Response:
We thank the reviewer for the suggestion. We have included the suggested references in the revised manuscript.
Comment (4) In the conclusion part, some prospects need to be added about magnetic nanometer catalyst for syngas production and conversion.
Response:
We have modified the conclusion section. We have included more prospects in the conclusion section.
Round 2
Reviewer 1 Report
very good
Reviewer 3 Report
The authors have carefully revised the manuscript(Manuscript ID: catalysts- 2219002)and made satisfactory response to me. I think the revised manuscript is suitable for publication in Catalysts at this state.